# MPAS-Albany Land Ice (MALI): A variable resolution ice sheet model for Earth system modeling using Voronoi grids

Matthew J. Hoffman[1], Mauro Perego[2], Stephen F. Price[1], William H. Lipscomb[1,3], Tong Zhang[1], Douglas Jacobsen[1], Irina Tezaur[4], Andrew G. Salinger[2], Raymond Tuminaro[2], and Luca Bertagna[2]

[1]Fluid Dynamics and Solid Mechanics Group, Los Alamos National Laboratory, P.O. Box 1663, MS B216, Los Alamos, NM, 87545, USA.
[2]Center for Computing Research, Sandia National Laboratories, P.O. Box 5800, MS 1320, Albuquerque, NM 87185, USA.
[3]National Center for Atmospheric Research, Boulder, CO, USA.
[4]Extreme Scale Data Science and Analytics, Sandia National Laboratories, P.O. Box 969, MS 9159, Livermore, CA 94551, USA.

*Correspondence to:* Matthew Hoffman (mhoffman@lanl.gov)

**Abstract.** We introduce MPAS-Albany Land Ice (MALI) v6.0, a new, variable resolution land ice model that uses unstructured Voronoi grids on a plane or sphere. MALI is built using the Model for Prediction Across Scales (MPAS) framework for developing variable resolution Earth System Model components and the Albany multi-physics code base for solution of coupled systems of partial-differential equations, which itself makes use of Trilinos solver libraries. MALI includes a three-dimensional, first-order momentum balance solver ("Blatter-Pattyn") by linking to the Albany-LI ice sheet velocity solver, as well as an explicit shallow ice velocity solver. Evolution of ice geometry and tracers is handled through an explicit first-order horizontal advection scheme with vertical remapping. Evolution of ice temperature is treated using operator splitting of vertical diffusion and horizontal advection and can be configured to use either a temperature or enthalpy formulation. MALI includes a mass-conserving subglacial hydrology model that supports distributed and/or channelized drainage and can optionally be coupled to ice dynamics. Options for calving include "eigencalving", which assumes calving rate is proportional to extensional strain rates. MALI is evaluated against commonly used exact solutions and community benchmark experiments and shows the expected accuracy. Results for the MISMIP3d benchmark experiments with MALI's Blatter-Pattyn solver fall in-between published results from Stokes and L1L2 models as expected. We use the model to simulate a semi-realistic Antarctic Ice Sheet problem following the initMIP protocol and using 2 km resolution in marine ice sheet regions. MALI is the glacier component of the Energy Exascale Earth System Model (E3SM) version 1, and we describe current and planned coupling to other E3SM components.

## 1 Introduction

During the past decade, numerical ice sheet models (ISMs) have undergone a renaissance relative to their predecessors. This period of intense model development was initiated following the Fourth Assessment Report of the Intergovernmental Panel on Climate Change (IPCC, 2007), which pointed to deficiencies in ISMs of the time as being the single largest shortcoming with respect to the scientific community's ability to project future sea-level rise stemming from ice sheets. Model maturation during

this period, which continued through the IPCC's Fifth Assessment Report (IPCC, 2013) and to the present day, has focused on improvements to ISM "dynamical cores" (including the fidelity, discretization, and solution methods for the governing conservation equations (e.g., Bueler and Brown, 2009; Schoof and Hindmarsh, 2010; Goldberg, 2011; Perego et al., 2012; Leng et al., 2012; Larour et al., 2012; Aschwanden et al., 2012; Cornford et al., 2013; Gagliardini et al., 2013; Brinkerhoff and Johnson, 2013)), ISM model "physics" (for example, the addition of improved models of basal sliding coupled to explicit subglacial hydrology (e.g., Schoof, 2005; Werder et al., 2013; Hewitt, 2013; Hoffman and Price, 2014; Bueler and van Pelt, 2015); and ice damage, fracture, and calving (e.g., Åström et al., 2014; Bassis and Ma, 2015; Borstad et al., 2016; Jiménez et al., 2017)) and the coupling between ISMs and Earth System Models (ESMs) (e.g., Ridley et al., 2005; Vizcaíno et al., 2008, 2009; Fyke et al., 2011; Lipscomb et al., 2013). These "next generation" ISMs have been applied to community-wide experiments focused on assessing (i) the sensitivity of ISMs to idealized and realistic boundary conditions and environmental forcing and (ii) the potential future contributions of ice sheets to sea-level rise (see e.g., Pattyn et al., 2013; Nowicki et al., 2013a, b; Bindschadler et al., 2013; Shannon et al., 2013; Edwards et al., 2014b).

While these efforts represent significant steps forward, next-generation ISMs continue to confront new challenges. These come about as a result of (1) applying ISMs to larger (whole-ice sheet), higher-resolution (regionally $O(1 \text{ km})$ or less), and more realistic problems, (2) adding new or improved sub-models of critical physical processes to ISMs, and (3) applying ISMs as partially or fully coupled components of ESMs. The first two challenges relate to maintaining adequate performance and robustness, as increased resolution and/or complexity have the potential to increase forward model cost and/or degrade solver reliability. The latter challenge relates to the added complexity and cost associated with optimization workflows, which are necessary for obtaining model initial conditions that are realistic and compatible with forcing from ESMs. These challenges argue for ISM development that specifically targets the following model features and capabilities:

1. parallel, scalable, and robust, linear and nonlinear solvers

2. variable and / or adaptive mesh resolution

3. computational kernels based on flexible programming models, to allow for implementation on a range of High-Performance Computing (HPC) architectures[1]

4. automatic differentiation capability for the computation of adjoint sensitivities to be used in high-dimensional parameter field optimization and uncertainty quantification

Based on these considerations, we have developed a new land ice model, the MALI model, which is composed of three major components: 1) model framework, 2) dynamical cores for solving equations of conservation of momentum, mass, and energy, and 3) modules for additional model physics. The model leverages existing and mature frameworks and libraries, namely the Model for Prediction Across Scales (MPAS) framework and the Albany and Trilinos solver libraries. These have

---

[1]For example, traditional CPU-only architectures and MPI programming models versus CPU+GPU, hybrid architectures using MPI for nodal communication and OpenMP or CUDA for on-node parallelism.

allowed us to take into consideration and address, from the start, many of the challenges discussed above. We discuss each of these components in more detail in the following sections.

## 2   MPAS Framework

The MPAS Framework provides the foundation for a generalized geophysical fluid dynamics model on unstructured spherical and planar meshes. On top of the framework, implementations specific to the modeling of a particular physical system (e.g., land ice, ocean) are created as MPAS *cores*. To date, MPAS cores for atmosphere (Skamarock et al., 2012), ocean (Ringler et al., 2013; Petersen et al., 2015, 2018), shallow water (Ringler et al., 2011), sea ice (Turner et al., 2018), and land ice have been implemented. At the moment the land ice model is limited to planar meshes due to the planar formulation of the flow models; however, we have an experimental implementation of the flow model for spherical coordinates that enables runs on spherical meshes. The MPAS design philosophy is to leverage the efforts of developers from the various MPAS cores to provide common framework functionality with minimal effort, allowing MPAS core developers to focus on development of the physics and features relevant to their application.

The framework code includes shared modules for fundamental model operation. Significant capabilities include:

– *Description of model data types.* MPAS uses a handful of fundamental Fortran derived types for basic model functionality. Model variables specific to an MPAS core are handled through custom groupings of model fields called *pools*, for which custom accessor routines exist. Core-specific variables are easily defined in XML syntax in a *Registry*, and the framework parses the Registry, defines variables, and allocates memory as needed.

– *Description of the mesh specification.* MPAS requires 36 fields to fully describe the mesh used in a simulation. These include the position, area, orientation, and connectivity of all cells, edges, and vertices in the mesh. The mesh specification can flexibly describe both spherical and planar meshes. More details are provided in the next section.

– *Distributed memory parallelization and domain decomposition.* The MPAS Framework provides needed routines for exchanging information between processors in a parallel environment using Message Passing Interface (MPI). This includes halo updates, global reductions, and global broadcasts. MPAS also supports decomposing multiple domain blocks on each processor to, for example, optimize model performance by minimizing transfer of data from disk to memory. Shared memory parallelization through OpenMP is also supported, but the implementation is left up to each MPAS core.

– *Parallel input and output capabilities.* MPAS performs parallel input and output of data from and to disk through the commonly used libraries of NetCDF, Parallel NetCDF (pnetcdf), and Parallel Input/Output (PIO) (Dennis et al., 2012). The Registry definitions control which fields can be input and/or output, and a framework *streams* functionality provides easy run-time configuration of what fields are to be written to what file name and at what frequency through an XML streams file. The MPAS framework includes additional functionality specific to providing a flexible model restart capability.

- *Advanced timekeeping.* MPAS uses a customized version of the timekeeping functionality of the Earth System Modeling Framework (ESMF), which includes a robust set of time and calendar tools used by many Earth System Models (ESMs). This allows explicit definition of model epochs in terms of years, months, days, hours, minutes, seconds, and fractional seconds and can be set to three different calendar types: Gregorian, Gregorian no leap, and 360 day. This flexibility helps enable multi-scale physics and simplifies coupling to ESMs. To manage the complex date/time types that ensue, MPAS framework provides routines for arithmetic of time intervals and the definition of alarm objects for handling events (e.g., when to write output, when the simulation should end).

- *Run-time configurable control of model options.* Model options are configured through *namelist* files that use standard Fortran namelist file format, and input/output are configured through *streams* files that use XML format. Both are completely adjustable at run time.

- *Online, run-time analysis framework.* See section Sect. 6.2 for examples.

Additionally, a number of shared operators exist to perform common operations on model data. These include geometric operations (e.g., length, area, and angle operations on the sphere or the plane), interpolation (linear, barycentric, Wachspress, radial basis functions, spline), vector and tensor operations (e.g., cross products, divergence), and vector reconstruction (e.g., interpolating from cell edges to cell centers). Most operators work on both spherical and planar meshes.

## 2.1 Model Meshes

The MPAS mesh specification is general enough to describe unstructured meshes on most two-dimensional manifold spaces, however most applications use centroidal Voronoi tesselations (Du and Gunzburger, 2002) on a sphere or plane. This manuscript focuses on applications with planar centroidal Voronoi meshes, with some additional consideration of spherical centroidal Voronoi meshes. Voronoi meshes are constructed by specifying a set of generating points (cell centers), and then partitioning the domain into cells that contain all points closer to each generating point than any other. Edges of Voronoi cells are equidistant between neighboring cell centers and perpendicular to the line connecting those cell centers. A planar Voronoi tesselation is the dual graph of a Delaunay triangulation, which is a triangulation of points in which the circumcircle of every triangle contains no points in the point set. Voronoi meshes that are centroidal (the Voronoi generator is also the center of mass of the cell) have favorable properties for some geophysical fluid dynamic applications (Ringler et al., 2010) and maintain high quality cells because cells tend towards equidimensional aspect ratios, and mesh resolution (where non-uniform) changes smoothly. On both planes and spheres, Voronoi tesselations tend toward perfect hexagons as resolution is increased. Note that, while the MPAS mesh specification supports quadrilateral grids, such as traditional rectangular grids, they are described as unstructured, which introduces significant overhead in memory and calculation over regular rectangular grid approaches.

Because MPAS meshes are two-dimensional manifold spaces, they are convenient for describing geophysical locations, either on planar projections or directly on a sphere. Because they are unstructured, meshes can contain varying mesh resolution and can be culled to only retain regions of interest. Planar meshes can easily be made periodic by taking advantage of the unstructured mesh specification and, for most operations, periodic cell relationships are handled the same as for neighboring

cell relationships. MPAS meshes are static in time. The vertical coordinate, if needed by an MPAS core, is extruded from the base horizontal mesh. Each MPAS core chooses its own vertical coordinate system. A comprehensive suite of tools for the generation of centroidal Voronoi tesselations on a plane or sphere has been developed, as well as tools for modifying existing meshes (e.g., removing unneeded cells, coordinate transformations, etc.) and converting some common unstructured mesh formats (e.g., *Triangle*, Shewchuk (1996)) to the MPAS specification.

The basic unit of the MPAS mesh specification is the *cell*. A cell has area and is formed by three or more sides, which are referred to as *edges*. The endpoints of edges are defined by *vertices*. Figure 1 illustrates the relationships between these mesh primitives. The MPAS mesh specification utilizes 36 fields that describe the position, orientation, area, and connectivity of the various primitives. Only four of these fields ($x, y, z$ cell positions and connectivity between cells) are necessary to describe any mesh, but the larger set of fields in the mesh specification provide information that is commonly used for routine operations. This avoids the need for the model to calculate these fields internally, speeding up the process of model initialization and integration.

MALI typically uses centroidal Voronoi meshes on a plane. Spherical Voronoi meshes can also be used, but little work has been done with such meshes to date. MALI employs a C-grid discretization (Arakawa and Lamb, 1977) for advection, meaning state variables (ice thickness and tracer values) are located at Voronoi cell centers, and flow variables (transport velocity, $u_n$) are located at cell edge midpoints (Fig. 1). MALI uses a sigma vertical coordinate (specified number of layers, each with a spatially uniform layer thickness fraction, see (Petersen et al., 2015) for more information):

$$\sigma = \frac{s - z}{H} \tag{1}$$

where $s$ is surface elevation, $H$ is ice thickness, and $z$ is the vertical coordinate.

A set of tools supporting the MPAS Framework includes tools for generating uniform and variable resolution centroidal Voronoi meshes. Additionally, the JIGSAW(GEO) mesh generation tool (Engwirda, 2017a, b) can be used to efficiently generate high quality, variable resolution meshes with data-based density functions. Density functions that are a function of observed ice velocity or its spatial derivatives and/or distance to the existing or potential future grounding line position have been used.

## 3  The Albany Software Library

Albany is an open source, C++ multi-physics code base for the solution and analysis of coupled systems of partial-differential equations (PDEs) (Salinger et al., 2016). It is a finite element code that can (in three spatial dimensions) employ unstructured meshed comprised of hexahedral, tetrahedral, or prismatic elements. Albany is designed to take advantage of the computational mathematics tools available within the Trilinos suite of software libraries (Heroux et al., 2005) and it uses template-based generic programming methods to provide extensibility and flexibility (Pawlowski et al., 2012). Together, Albany and Trilinos provide parallel data structures and I/O, discretization and integration algorithms, linear solvers and preconditioners, nonlinear solvers, continuation algorithms, and tools for automatic differentiation (AD) and optimization. By formulating a system of equations in the residual form, Albany employs AD to automatically compute the Jacobian of the discrete PDE residual, as

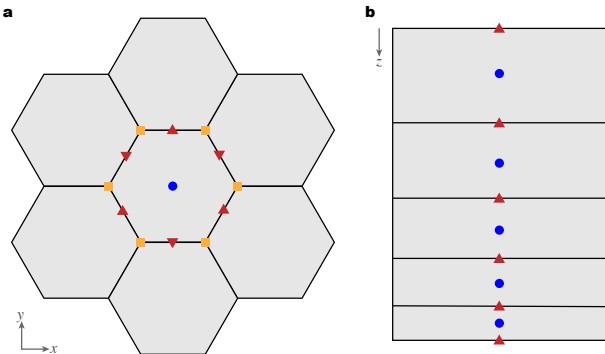

**Figure 1.** MALI grids. a) Horizontal grid with cell center (blue circles), edge midpoint (red triangles), and vertices (orange squares) identified for the center cell. Scalar fields $(H, T)$ are located at cell centers. Advective velocities $(u_n)$ and fluxes are located at cell edges. b) Vertical grid with layer midpoints (blue circles) and layer interfaces (red triangles) identified. Scalar fields $(H, T)$ are located at layer midpoints. Fluxes are located at layer interfaces.

well as forward and adjoint sensitivities. Albany can solve large-scale PDE-constrained optimization problems, using Trilinos optimization package ROL, and it provides uncertainty quantification capabilities through the Dakota framework (Adams et al., 2013). It is a massively parallel code by design and recently it has been adopting the Kokkos (Edwards et al., 2014a) programming model to provide manycore performance portability (Demeshko et al., 2018) on major HPC platforms. Albany provides several applications including LCM (Laboratory for Computational Mechanics) for solid mechanics problems, QCAD (Quantum Computer Aided Design) for quantum device modeling, and LI (Land Ice) for modeling ice sheet flow. We refer to the code that discretizes these diagnostic momentum balance equations as Albany-LI. Albany-LI was formerly known as Albany/FELIX (Finite Elements for Land Ice eXperiments), and described by Tezaur et al. (2015a, b) and Tuminaro et al. (2016) under that name. Here, these tools are brought to bear on the most complex, expensive, and fragile portion of the ice sheet model, the solution of the momentum balance equations (discussed further below).

## 4   Conservation Equations

The "dynamical core" of the the MALI ice sheet model solves the governing equations expressing the conservation of momentum, mass, and energy.

### 4.1   Conservation of Momentum

Treating glacier ice as an incompressible fluid in a low-Reynolds number flow, the conservation of momentum in a Cartesian reference frame is expressed by the Stokes-flow equations, for which the gravitational-driving stress is balanced by gradients

in the viscous stress tensor, $\sigma_{ij}$:

$$\frac{\partial \sigma_{ij}}{\partial x_j} + \rho g_i = 0, \ \ i,j = 1,2,3 \tag{2}$$

where $x_i$ is the coordinate vector, $\rho$ is the density of ice, and $g =$ is acceleration due to gravity[2].

Deformation results from the deviatoric stress, $\tau_{ij}$, which relates to the full stress tensor as

$$\tau_{ij} = \sigma_{ij} - \frac{1}{3}\sigma_{kk}\delta_{ij}, \tag{3}$$

for which $-\frac{1}{3}\sigma_{kk}$ is the mean compressive stress and $\delta_{ij}$ is the Kroneker delta (or the identity tensor). Stress and strain rate are related through the constitutive relation,

$$\tau_{ij} = 2\mu_e \dot{\epsilon}_{ij}, \tag{4}$$

where $\dot{\epsilon}_{ij}$ is the strain-rate tensor and $\mu_e$ is the "effective", non-Newtonian ice viscosity given by Nye's generalization of Glen's flow law (Glen, 1955),

$$\mu_e = \gamma A^{-\frac{1}{n}} \dot{\epsilon}_e^{\frac{1-n}{n}}. \tag{5}$$

In Eq. (5), $A$ is a temperature dependent rate factor, $n$ is an exponent commonly taken as 3 for polycrystalline glacier ice, and $\gamma$ is an ice "stiffness" factor (inverse enhancement factor related to the commonly used enhancement factor $E_f$ by $\gamma = E_f^{\frac{-1}{n}}$) used to account for other impacts on ice rheology, such as impurities or crystal anisotropy (see also Sect. 6.1). The effective strain rate $\dot{\epsilon}_e$ is given by the second invariant of the strain-rate tensor,

$$\dot{\epsilon}_e = \left(\frac{1}{2}\dot{\epsilon}_{ij}\dot{\epsilon}_{ij}\right)^{\frac{1}{2}}, \tag{6}$$

The strain rate tensor is defined by gradients in the components of the ice velocity vector $u_i$:

$$\dot{\epsilon}_{ij} = \frac{1}{2}\left(\frac{\partial u_i}{\partial x_j} + \frac{\partial u_j}{\partial x_i}\right), \ \ i,j = 1,2,3. \tag{7}$$

Finally, the rate-factor $A$ follows an Arrhenius relationship

$$A(T^*) = A_o e^{-Q_a/RT^*}, \tag{8}$$

in which $A_o$ is a constant, $T^*$ is the temperature (relative to the pressure melting point), $Q_a$ is the activation energy for crystal creep, and $R$ is the gas constant.

Boundary conditions required for the solution of Eq. (2) depend on the form of reduced-order approximation applied and are discussed further below.

---

[2]In Eq. (2) and elsewhere we use indicial notation, with summation over repeat indices.

## 4.2 Reduced-order Equations

Ice sheet models solve Eqs. (2)-(8) with varying degrees of complexity in terms of the tensor components in Eqs. (2)-(7) that are accounted for or omitted, based on geometric scaling arguments. Because ice sheets are inherently "thin" – their widths are several orders of magnitude larger than their thickness – reduced-order approximations of the full momentum balance are often appropriate (see, e.g., Dukowicz et al., 2010; Schoof and Hewitt, 2013) and, importantly, can often result in considerable computational cost savings. Here, we employ two such approximations, a first-order-accurate "Blatter-Pattyn" approximation and a zero-order, "shallow-ice approximation" as described in more detail in the following sections.

### 4.2.1 First-Order Velocity Solver and Coupling

Ice sheets typically have a small aspect ratio and small surface and bed slopes. These characteristics imply that reduced-order approximations of the Stokes momentum balance may apply over large areas of the ice sheets, potentially allowing for significant computational savings. Formal derivations involve non-dimensionalizing the Stokes momentum balance and introducing a geometric scaling factor, $\delta = H/L$, where $H$ and $L$ represent characteristic vertical and horizontal length scales (often taken as the ice thickness and the ice sheet span), respectively. Upon conducting an asymptotic expansion, reduced-order models with a chosen degree of accuracy (relative to the original Stokes flow equations) can be derived by retaining terms of the appropriate order in $\delta$. For example, the first-order accurate Stokes approximation is arrived at by retaining terms of $\mathcal{O}(\delta^1)$ and lower (The reader is referred to Schoof and Hindmarsh (2010) and Dukowicz et al. (2010) for additional discussion[3]).

Using the notation of Perego et al. (2012) and Tezaur et al. (2015a) [4], the first-order accurate Stokes approximation (also referred to as the "Blatter-Pattyn" approximation, see Blatter, 1995; Pattyn, 2003) is expressed through the following system of PDEs,

$$\begin{cases} -\nabla \cdot (2\mu_e \dot{\boldsymbol{\epsilon}}_1) + \rho g \frac{\partial s}{\partial x} &= 0, \\ -\nabla \cdot (2\mu_e \dot{\boldsymbol{\epsilon}}_2) + \rho g \frac{\partial s}{\partial y} &= 0, \end{cases} \tag{9}$$

where $\nabla \cdot$ is the divergence operator, $s \equiv s(x,y)$ represents the ice sheet upper surface, and the vectors $\dot{\boldsymbol{\epsilon}}_1$ and $\dot{\boldsymbol{\epsilon}}_2$ are given by

$$\dot{\boldsymbol{\epsilon}}_1 = \left(\begin{array}{ccc} 2\dot{\epsilon}_{xx} + \dot{\epsilon}_{yy}, & \dot{\epsilon}_{xy}, & \dot{\epsilon}_{xz} \end{array}\right)^T, \tag{10}$$

and

$$\dot{\boldsymbol{\epsilon}}_2 = \left(\begin{array}{ccc} \dot{\epsilon}_{xy}, & \dot{\epsilon}_{xx} + 2\dot{\epsilon}_{yy}, & \dot{\epsilon}_{yz} \end{array}\right)^T. \tag{11}$$

Akin to Eqs. (5) and (6), $\mu_e$ in Eq. (9) represents the effective viscosity but for the case of the first-order stress balance with an effective-strain rate is given by

$$\dot{\epsilon}_e \equiv \left(\dot{\epsilon}_{xx}^2 + \dot{\epsilon}_{yy}^2 + \dot{\epsilon}_{xx}\dot{\epsilon}_{yy} + \dot{\epsilon}_{xy}^2 + \dot{\epsilon}_{xz}^2 + \dot{\epsilon}_{yz}^2\right)^{\frac{1}{2}}, \tag{12}$$

---

[3]In practice, additional scaling parameters describing the ratio of deformation to sliding velocity may also be introduced.

[4]Vectors and tensors are given in bold rather than using indices. Note that, in a slight abuse of notation, we have switched from using $x_1$, $x_2$, $x_3$ to denote the three coordinate directions to $x$, $y$, $z$.

rather than by Eq. (6), and with individual strain rate terms given by,

$$\dot{\epsilon}_{xx} = \frac{\partial u}{\partial x}, \quad \dot{\epsilon}_{yy} = \frac{\partial v}{\partial y}, \quad \dot{\epsilon}_{xy} = \frac{1}{2}\left(\frac{\partial u}{\partial y} + \frac{\partial v}{\partial x}\right), \quad \dot{\epsilon}_{xz} = \frac{1}{2}\frac{\partial u}{\partial z}, \quad \dot{\epsilon}_{yz} = \frac{1}{2}\frac{\partial v}{\partial z}. \tag{13}$$

At the upper surface, a stress-free boundary condition is applied,

$$\dot{\boldsymbol{\epsilon}}_1 \cdot \mathbf{n} = \dot{\boldsymbol{\epsilon}}_2 \cdot \mathbf{n} = 0, \tag{14}$$

with $\mathbf{n}$ the outward normal vector at the ice sheet surface, $z = s(x,y)$. At the bed, $z = b(x,y)$, we apply no slip or continuity of basal tractions ("sliding"),

$$\begin{aligned} u = v = 0, & \qquad\qquad \text{no slip} \\ 2\mu_e\dot{\boldsymbol{\epsilon}}_1 \cdot \mathbf{n} + \beta u^m = 0, \;\; 2\mu_e\dot{\boldsymbol{\epsilon}}_2 \cdot \mathbf{n} + \beta v^m = 0, & \quad \text{sliding,} \end{aligned} \tag{15}$$

where $\beta$ is a linear-friction parameter and $m \geq 1$. In most applications we set $m = 1$ (see also Sect. 5.1.6).

On lateral boundaries, a stress boundary condition is applied,

$$2\mu_e\left(\dot{\boldsymbol{\epsilon}}_1 \cdot \mathbf{n}, \dot{\boldsymbol{\epsilon}}_2 \cdot \mathbf{n}, 0\right)^T - \rho g(s - z)\mathbf{n} = \rho_o g \max(z, 0)\mathbf{n}, \tag{16}$$

where $\rho_o$ is the density of ocean water and $\mathbf{n}$ the outward normal vector to the lateral boundary (i.e., parallel to the $(x,y)$ plane), so that lateral boundaries above sea level are effectively stress free and lateral boundaries submerged within the ocean experience hydrostatic pressure due to the overlying column of ocean water.

We solve these equations using the Albany-LI momentum balance solver, which is built using the Albany and Trilinos
software libraries discussed above. The mathematical formulation, discretization, solution methods, verification, and scaling of Albany-LI are discussed in detail in Tezaur et al. (2015a). Albany-LI implements a classic finite element discretization of the first-order approximation. At the grounding line, the basal friction coefficient $\beta$ can abruptly drop to zero within an element of the mesh. This discontinuity is resolved by using an higher-order Gauss quadrature rule on elements containing the grounding line, which corresponds to the sub-element parametrization *SEP3* proposed in Seroussi et al. (2014). Additional
exploration of solver scalability and demonstrations of solver robustness on large scale, high-resolution, realistic problems are discussed in Tezaur et al. (2015b). The efficiency and robustness of the nonlinear solvers are achieved using a combination of the Newton method (damped with a line search strategy when needed) and of a parameter continuation algorithm for the numerical regularization of the viscosity. The scalability of the linear solvers is obtained using a multilevel preconditioner (see Tuminaro et al. (2016)) specifically designed to target shallow problems characterized by meshes extruded in the vertical
dimension, like those found in ice sheet modeling. The preconditioner has been demonstrated to be particularly effective and robust even in the presence of ice shelves that typically lead to highly ill-conditioned linear systems.

The Albany-LI first-order velocity solver written in C++ is coupled to MPAS written in Fortran using an interface layer. Albany uses a three-dimensional mesh extruded from a basal triangulation and composed of prisms or tetrahedra (see Tezaur et al. (2015a)). When coupled to MPAS, the basal triangulation is the part of the Delaunay triangulation, dual to an MPAS
Voronoi mesh, that contains active ice and it is generated by the interface. Bed topography, ice lower surface, ice thickness,

**Table 1.** Correspondence between the MPAS Voronoi tesselation and its dual Delaunay triangulation used by Albany. Key MALI model variables that are natively found at each location are listed. Note that variables are interpolated from one location to another as required for various calculations.

| Voronoi tesselation | Delaunay triangulation | Variables |
|---|---|---|
| cell center | triangle node | $H, T, u, v, \Phi$ (MPAS) |
| cell edge | triangle edge | $u_n$ (for advection) |
| cell vertex | triangle center | $\Phi$ (Albany) |

basal friction coefficient ($\beta$), and three-dimensional ice temperature, all at cell centers (Table 1), are passed from MPAS to Albany. Optionally, Dirichlet velocity boundary conditions can also be passed. After the velocity solve is complete, Albany returns the $x$ and $y$ components of velocity at each cell center and layer interfaces, the normal component of velocity at each cell edge and layer interfaces, and viscous dissipation at each cell vertex and layer midpoints.

The interface code defines the lateral boundary conditions on the finite element mesh that Albany will use. Lateral boundaries in Albany are applied at cell centers (triangle nodes) that do not contain dynamic ice on the MPAS mesh and that are adjacent to the last cell of the MPAS mesh that does contain dynamic ice. This one element extension is required to support calculation of normal velocity on edges ($u_n$) required for advection of ice out of the final cell containing dynamic ice (Fig. 2). The interface identifies three types of lateral boundaries for the first-order velocity solve: terrestrial, floating marine, and grounded

marine. Terrestrial margins are defined by bed topography above sea level. At these boundary nodes, ice thickness is set to a small ice minimum thickness value ($\epsilon = 1$ m). Floating marine margin triangle nodes are defined as neighboring one or more triangle edges that satisfy the hydrostatic floatation criterion. At these boundary nodes, we need to ensure the existence of a realistic calving front geometry, so we set ice thickness to the minimum of thickness at neighboring cells with ice. Grounded marine margins are defined as locations where the bed topography is below sea level, but no adjacent triangle edges satisfy

the floatation criterion. At these boundary nodes, we apply a small floating extension with thickness $\epsilon$. For all three boundary types, ice temperature is averaged from the neighboring locations containing ice.

### 4.2.2   Shallow-Ice Approximation Velocity Solver

A similar procedure to that described above for the first-order accurate Stokes approximation can be used to derive the so-called "shallow-ice approximation" (SIA) (Hutter, 1983; Fowler and Larson, 1978; Morland and Johnson, 1980; Payne et al.,

2000), in this case by retaining only terms of $\mathcal{O}(\delta^0)$. In the case of the SIA, the local gravitational driving stress is everywhere balanced by the local basal traction and the horizontal velocity as a function of depth is simply the superposition of the local basal sliding velocity and the integral of the vertical shear from the ice base to that depth:

$$\boldsymbol{u} = -2(\rho g)^n \left( \int_b^z A(s-z)^n dz \right) |\nabla s|^{n-1} \nabla s + \boldsymbol{u_b} \tag{17}$$

where $b$ is the bed elevation and $\boldsymbol{u_b}$ is the sliding velocity.

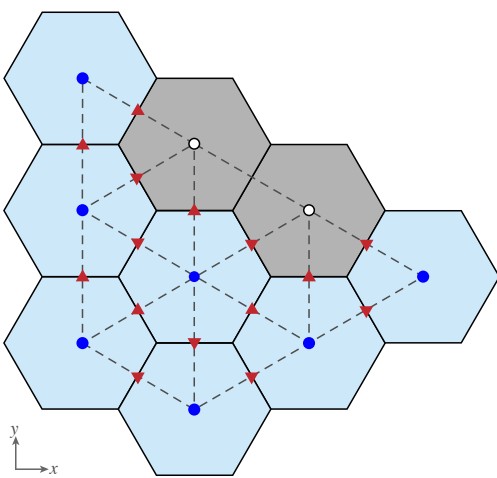

**Figure 2.** Correspondence between MPAS and Albany meshes and application of boundary conditions for the first-order velocity solver. Solid black lines are cells on the Voronoi mesh and dashed gray lines are triangles on the Delaunay Triangulation. Light blue Voronoi cells contain dynamic ice and gray cells do not. Dark blue circles are Albany triangle nodes that use variable values directly from the co-located MPAS cell centers. White circles are extended node locations that receive variable values as described in the text based on whether they are terrestrial, floating marine, or grounded marine locations. Red triangles indicate Voronoi cell edges on which velocities ($u_n$) are required for advection.

SIA ice sheet models typically combine the momentum and mass balance equations to evolve the ice geometry directly in the form of a depth-integrated, two-dimensional diffusion problem (Hindmarsh and Payne, 1996; Payne et al., 2000). However we implement the SIA as an explicit velocity solver that can be enabled in place of the more accurate first order solver, while keeping the rest of the model identical. The purpose of the SIA velocity solver is primarily for rapid testing, so the less efficient
explicit implementation of Eq. (17) is not a concern.

We implement Eq. (17) in sigma coordinates on cell edges, where we only require the normal component of velocity, $u_n$:

$$u_n = -2(\rho g)^n H^{n+1} |\nabla s|^{n-1} \frac{ds}{dx_n} \int_1^\sigma A\sigma^n d\sigma + u_{b_n} \tag{18}$$

where $x_n$ is the normal direction to a given edge and $u_{b_n}$ is sliding velocity in the normal direction to the edge. We average $A$ and $H$ from cell centers to cell edges. $\frac{ds}{dx_n}$ is calculated as the difference in surface elevation between the two cells that
neighbor a given edge divided by the distance between the cell centers; on a Voronoi grid, cells edges are midway between cell centers by definition. The surface slope component tangent to an edge (required to complete the calculation of $\nabla s$) is calculated by first interpolating surface elevation from cell centers to vertices.

### 4.3    Conservation of Mass

Ice sheet mass transport and evolution is conducted using the principle of conservation of mass. Assuming constant density to
write conservation of mass in volume form, the equation relates ice thickness change to the divergence of mass and sources

and sinks:

$$\frac{\partial H}{\partial t} + \nabla \cdot H\overline{\mathbf{u}} = \dot{a} + \dot{b}, \tag{19}$$

where $H$ is ice thickness, $t$ is time, $\overline{\mathbf{u}}$ is depth-averaged velocity, $\dot{a}$ is surface mass balance, and $\dot{b}$ is basal mass balance. Both $\dot{a}$ and $\dot{b}$ are positive for ablation and negative for accumulation.

Eq. (19) is used to update thickness in each grid cell on each time step using a forward Euler, fully explicit time evolution scheme. Eq. (19) is implemented using a finite volume method, such that fluxes are calculated for each edge of each cell to calculate $\nabla \cdot H\overline{\mathbf{u}}$. Specifically, we use a first-order upwind method that applies the normal velocity on each edge ($u_n$) and an upwind value of cell centered ice thickness. Note that with the First Order velocity solver, normal velocity is interpolated from cell centers to edges using the finite element basis functions in Albany. In the shallow ice approximation velocity solver, normal velocity is calculated natively at edges. MPAS Framework includes a higher-order flux-corrected transport scheme (Ringler et al., 2013) for which we have performed some initial testing, but is not routinely used in MALI at this time.

Tracers are advected horizontally layer-by-layer with a similar equation:

$$\frac{\partial (Q_t l)}{\partial t} + \nabla \cdot (Q_t l \overline{\mathbf{u}}) = \dot{S} \tag{20}$$

where $Q_t$ is a tracer quantity (e.g., temperature – see below), $l$ is layer thickness, and $\dot{S}$ represents any tracer sources or sinks. While any number of tracers can be included in the model, the only one to be considered here is temperature, due to its important effect on ice rheology through Eq. (8) and will be discussed further in the following section.

Vertical advection of tracers is included through a vertical remapping operation. On the upper and lower domain boundaries, the grid moves to follow the material, and in the interior we maintain fixed layer fractions that need to be updated on each time step after Eqs. (19) and (20) are applied. The model does not explicitly calculate vertical velocity, but the appropriate vertical transport of tracers occurs during this vertical remapping operation. We employ a first-order vertical remapping method. Overlaps between the newly calculated layers and the target sigma layers are calculated for each grid cell. Assuming uniform values within each layer, mass, energy, and other tracers are transferred between layers based on these overlaps to restore the prescribed sigma layers while conserving mass and energy.

## 4.4 Conservation of Energy

Conservation of energy within glaciers can be formulated in terms of temperature or enthalpy (internal energy) (Aschwanden et al., 2012; Kleiner et al., 2015). The enthalpy formulation has the advantage of eliminating the need of tracking the cold-temperate transition surface, as both cold (below the pressure melting point) and temperate (at the pressure melting point) ice regions are handled with the same equations. MALI includes both temperature and enthalpy formulations. In both cases, an operator splitting technique is used. At each time step, an implicit vertical solve accounting for the diffusion and dissipation terms (described below) is performed, followed by explicit advection of the resulting temperature or enthalpy field (described above in Sect. 4.3). We describe the temperature formulation in detail, followed by a briefer description of the enthalpy formulation that uses a similar procedure. Note that the thermal model described here shares a common lineage with that of the Community

Ice Sheet Model, and parts of the description below therefore are similar to the documentation of the thermal solver in the Community Ice Sheet Model (Price et al., 2015; Lipscomb et al., 2018).

### 4.4.1 Temperature Formulation

Conservation of energy can be expressed in terms of temperature through the three-dimensional, advective-diffusive heat equation:

$$\frac{\partial T}{\partial t} = \frac{1}{\rho c}\frac{\partial}{\partial x_i}\left(k\frac{\partial T}{\partial x_i}\right) - u_i\frac{\partial T}{\partial x_i} + \frac{\Phi}{\rho c}, \tag{21}$$

with thermal conductivity $k$ and heat capacity $c$. In Eq. (21), the rate of temperature change (left-hand side) is balanced by diffusive, advective, and internal (viscous dissipation – see Eq. (28) for $\Phi$) source terms (first, second, and third terms on the right-hand side, respectively). In MALI we solve an approximation of Eq. (21),

$$\frac{\partial T}{\partial t} = \frac{k}{\rho c}\frac{\partial^2 T}{\partial z^2} - u_i\frac{\partial T}{\partial x_i} + \frac{\Phi}{\rho c}, \tag{22}$$

in which horizontal diffusion is assumed negligible (van der Veen, 2013, p. 280) and $k$ is assumed constant and uniform. The viscous dissipation term $\Phi$ is discussed further below.

Temperatures are staggered in the vertical relative to velocities and are located at the centers of $nz-1$ vertical layers, which are bounded by $nz$ vertical levels (grid point locations). This convention allows for conservative temperature advection, since the total internal energy in a column (the sum of $\rho c T \Delta z$ over $nz-1$ layers) is conserved under transport. The upper surface temperature $T_s$ and the lower surface temperature $T_b$, coincident with the surface and bed grid points, give a total of $nz+1$ temperature values within each column.

As mentioned above, Eq. (22) is solved by first performing an implicit vertical solve accounting for the diffusion and dissipation terms (described below), followed by explicit advection of the resulting temperature field. The method for evolving ice temperature and default parameter value choices are adapted from the implementation in the Community Ice Sheet Model (Price et al., 2015; Lipscomb et al., 2018), which is in turn based on the Glimmer model (Rutt et al., 2009). The choice of constant $k$ with a temperate ice value (Table A1) will lead to underestimation of conduction in cold ice. Relaxation of this assumption is planned for future releases of MALI.

**Vertical Diffusion**

Using a "sigma" vertical coordinate, the vertical diffusion portion of Eq. (22) can be discretized as:

$$\frac{\partial^2 T}{\partial z^2} = \frac{1}{H^2}\frac{\partial^2 T}{\partial \sigma^2}. \tag{23}$$

In $\sigma$–coordinates, the central difference formulas for first partial derivatives at the upper and lower interfaces of layer $k$ are

$$\left.\frac{\partial T}{\partial \sigma}\right|_{\sigma_k} = \frac{T_k - T_{k-1}}{\tilde{\sigma}_k - \tilde{\sigma}_{k-1}},$$
$$\left.\frac{\partial T}{\partial \sigma}\right|_{\sigma_{k+1}} = \frac{T_{k+1} - T_k}{\tilde{\sigma}_{k+1} - \tilde{\sigma}_k}, \tag{24}$$

where $\tilde{\sigma}_k$ is the value of $\sigma$ at the midpoint of layer $k$, halfway between $\sigma_k$ and $\sigma_{k+1}$. The second partial derivative, defined at the midpoint of layer $k$, is then given by

$$\frac{\partial^2 T}{\partial \sigma^2}\bigg|_{\tilde{\sigma}_k} = \frac{\frac{\partial T}{\partial \sigma}\big|_{\sigma_{k+1}} - \frac{\partial T}{\partial \sigma}\big|_{\sigma_k}}{\sigma_{k+1} - \sigma_k} \tag{25}$$

By inserting Eq. (24) into Eq. (25), we obtain the discrete form of the vertical diffusion term in Eq. (22):

$$\frac{\partial^2 T}{\partial \sigma^2}\bigg|_{\tilde{\sigma}_k} = \frac{T_{k-1}}{(\tilde{\sigma}_k - \tilde{\sigma}_{k-1})(\sigma_{k+1} - \sigma_k)} - T_k \left( \frac{1}{(\tilde{\sigma}_k - \tilde{\sigma}_{k-1})(\sigma_{k+1} - \sigma_k)} + \frac{1}{(\tilde{\sigma}_{k+1} - \tilde{\sigma}_k)(\sigma_{k+1} - \sigma_k)} \right)$$
$$+ \frac{T_{k+1}}{(\tilde{\sigma}_{k+1} - \tilde{\sigma}_k)(\sigma_{k+1} - \sigma_k)}. \tag{26}$$

To simplify some expressions below, we define the following coefficients associated with the vertical temperature diffusion,

$$a_k = \frac{1}{(\tilde{\sigma}_k - \tilde{\sigma}_{k-1})(\sigma_{k+1} - \sigma_k)}, b_k = \frac{1}{(\tilde{\sigma}_{k+1} - \tilde{\sigma}_k)(\sigma_{k+1} - \sigma_k)}. \tag{27}$$

**Viscous Dissipation**

The source term from viscous dissipation in Eq. (22) is given by the product of the stress and strain rate tensors:

$$\Phi = \sigma_{ij}\dot{\epsilon}_{ij} = \tau_{ij}\dot{\epsilon}_{ij}. \tag{28}$$

The change to deviatoric stress on the right-hand side of Eq. (28) follows from terms related to the mean compressive stress (or pressure) dropping out due to incompressibility. Analogous to the effective strain rate given in Eq. (6), the effective-deviatoric stress is given by

$$\tau_e = \left( \frac{1}{2}\tau_{ij}\tau_{ij} \right)^{\frac{1}{2}}, \tag{29}$$

which can be combined with Eqs. (28) and (6) to derive an expression for the viscous dissipation in terms of effective deviatoric stress and strain,

$$\Phi = 2\tau_e\dot{\epsilon}_e \tag{30}$$

Finally, an analog to Eq. (4) gives

$$\tau_e = 2\mu_e\dot{\epsilon}_e, \tag{31}$$

which can be used to eliminate $\dot{\epsilon}_e$ in Eq. (30) and arrive at an alternate expression for the dissipation based on only two scalar quantities

$$\Phi = 4\mu_e\dot{\epsilon}_e^2. \tag{32}$$

The viscous dissipation source term is computed within Albany-LI at MPAS cell vertices and then reconstructed at cell centers in MPAS.

For the SIA model, dissipation can be calculated in sigma coordinates as

$$\Phi(\sigma) = \frac{\sigma g}{c} \frac{\partial \boldsymbol{u}}{\partial \sigma} \cdot \nabla s \tag{33}$$

which can be combined with Eq. (17) to make:

$$\Phi(\sigma) = -\frac{2\sigma g}{c\rho}(g\sigma\rho)^{n+1}(H|\nabla s|)^{n+1}A \tag{34}$$

We calculate $\Phi$ on cell edges following the procedure described for Eq. (18), and then interpolate $\Phi$ back to cell centers to solve Eq. (22).

**Vertical Temperature Solution**

The vertical diffusion portion of Eq. (22) is discretized according to

$$\frac{T_k^{n+1} - T_k^n}{\Delta t} = \frac{k}{\rho c H^2}\left(a_k T_{k-1}^{n+1} - (a_k + b_k)T_k^{n+1} + b_k T_{k+1}^{n+1}\right) + \frac{\Phi_k}{\rho c}, \tag{35}$$

where $a_k$ and $b_k$ are defined in Eq. (27), $n$ is the current time level, and $n+1$ is the new time level. Because the vertical diffusion terms are evaluated at the new time level, the discretization is backward-Euler (fully implicit) in time.

The temperature $T_0$ at the upper boundary is set to $\min(T_{\mathrm{air}}, 0)$, where the mean-annual surface air temperature $T_{\mathrm{air}}$ is a two-dimensional field specified from observations or climate model output.

At the lower boundary, for grounded ice there are three potential heat sources and sinks: (1) the diffusive flux from the bottom surface to the ice interior (positive up),

$$F_d^{\mathrm{bot}} = \frac{k}{H}\frac{T_{nz} - T_{nz-1}}{1 - \tilde{\sigma}_{nz-1}}; \tag{36}$$

(2) the geothermal flux $F_g$, prescribed from a spatially variable input file (based on observations), and (3) the frictional heat flux associated with basal sliding,

$$F_f = \tau_{\mathbf{b}} \cdot \mathbf{u_b}, \tag{37}$$

where $\tau_{\mathbf{b}}$ and $\mathbf{u_b}$ are 2D vectors of basal shear stress and basal velocity, respectively, and the friction law from Eq. (15) becomes

$$F_f = \beta\sqrt{u_b^2 + v_b^2}. \tag{38}$$

If the basal temperature $T_{nz} < T_{\mathrm{pmp}}$ (where $T_{\mathrm{pmp}}$ is the pressure melting point temperature), then the fluxes at the lower boundary must balance,

$$F_g + F_f = F_d^{\mathrm{bot}}, \tag{39}$$

so that the energy supplied by geothermal heating and sliding friction is equal to the energy removed by vertical diffusion. If, on the other hand, $T_{nz} = T_{\mathrm{pmp}}$, then the net flux is nonzero and is used to melt or freeze ice at the boundary:

$$M_b = \frac{F_g + F_f - F_d^{\mathrm{bot}}}{\rho L}, \tag{40}$$

where $M_b$ is the melt rate and $L$ is the latent heat of melting. Melting generates basal water, which may either be stored at the bed locally, serve as a source for the basal hydrology model (See Sect. 5.1), or may simply be ignored. If basal water is present locally, $T_{nz}$ is held at $T_{\text{pmp}}$.

For floating ice the basal boundary condition is simpler: $T_{nz}$ is simply set to the freezing temperature $T_f$ of seawater. Optionally, a melt rate can be prescribed at the lower surface.

Rarely, the solution for $T$ may exceed $T_{\text{pmp}}$ for a given internal layer. In this case, $T$ is set to $T_{\text{pmp}}$, excess energy goes towards melting of ice internally, and the resulting melt is assumed to drain to the bed immediately.

If Eq. 40 applies, we compute $M_b$ and adjust the basal water depth. When the basal water goes to zero, $T_{nz}$ is set to the temperature of the lowest layer (less than $T_{\text{pmp}}$ at the bed) and flux boundary conditions apply during the next time step.

**Temperature Advection**

Temperature advection in any individual layer $k$ is treated using tracer advection, as in Eq. (20) above, where the ice temperature $T_k$ is substituted for the generic tracer $Q$. After horizontal transport, the surface and basal mass balance is applied to the top and bottom ice surfaces, respectively. Because layer transport and the application of mass balance terms results in an altered vertical-layer spacing with respect to $\sigma$ coordinates, a vertical remapping scheme is applied to provide the necessary vertical advection of temperature. This conservatively transfers ice volume and internal energy between adjacent layers while restoring $\sigma$ layers to their initial distribution. Internal energy divided by mass gives the new layer temperatures.

### 4.4.2 Enthalpy Formulation

The specific enthalpy (internal energy), $E$, in ice sheets and glaciers can be expressed as a combination of ice temperature ($T$) and liquid water fraction (water content; $\omega$) (Aschwanden et al., 2012)

$$E = \begin{cases} c\,(T - T_{\text{ref}}), & E \leq E_{\text{pmp}} \\ E_{\text{pmp}} + \omega L, & E \geq E_{\text{pmp}} \end{cases} \tag{41}$$

where $T_{\text{ref}}$ is the reference temperature, $c$ is the heat capacity of ice and $L$ is the latent heat of fusion, and $E_{\text{pmp}}$ is the specific enthalpy at the pressure-melting-point for different vertical locations ($T_{\text{pmp}}(z)$), defined as

$$E_{\text{pmp}} = c\,(T_{\text{pmp}}(z) - T_{\text{ref}}) \tag{42}$$

The balance equation for enthalpy reads:

$$\frac{\partial E}{\partial t} = \frac{\partial}{\partial x_i}\left(K \frac{\partial E}{\partial x_i}\right) - u_i \frac{\partial E}{\partial x_i} + \Phi \tag{43}$$

where $K$ is the diffusivity of ice, defined differently in cold and temperate ice,

$$K = \begin{cases} \frac{k}{\rho c}, & E \leq E_{\text{pmp}} \\ \frac{\nu}{\rho}, & E \geq E_{\text{pmp}} \end{cases} \tag{44}$$

where $\nu$ is the water diffusivity in temperate ice, which is generally taken as an empirical small number due to a lack of its knowledge (Greve and Blatter, 2016).

The implementation of the enthalpy model follows that of the temperature model. Vertical diffusion is as described above but replacing $T$ with $E$. Viscous dissipation remains unchanged. Boundary conditions in the vertical enthalpy solution follow those applied for the temperature formulation above, but cast in terms of $E$. Advection is as described above for temperature but replacing $T$ with $E$. Verification of the enthalpy model is described below in Sect. 7.3.

## 5   Additional Model Physics

Additional physical processes currently implemented in MALI are a mass-conserving subglacial hydrology model and a small number of basic schemes for iceberg calving. These are described in more detail below.

### 5.1   Subglacial Hydrology

Sliding of glaciers and ice sheets over their bed can increase ice velocity by orders of magnitude and is the primary control on ice flux to the oceans. The state of the subglacial hydrologic system is the primary control on sliding (Clarke, 2005; Cuffey and Paterson, 2010; Flowers, 2015), and ice sheet modelers have therefore emphasized subglacial hydrology and its effects on basal sliding as a critical missing piece of current ice sheet models (Little et al., 2007; Price et al., 2011).

MALI includes a mass-conserving model of subglacial hydrology that includes representations of any or all of water storage in till, distributed drainage, and channelized drainage and is coupled to ice dynamics. The model is based on the model of Bueler and van Pelt (2015) with an additional component for channelized drainage and modified for MALI's unstructured horizontal grid. While the implementation follows closely that of Bueler and van Pelt (2015), the model and equations are summarized here along with a description of the features unique to the application in MALI.

#### 5.1.1   Till

The simple till component represents local storage of water in subglacial till without horizontal transport within the till. Evolution of the effective water depth in till, $W_{till}$ is therefore a balance of delivery of meltwater, $m_b$, to the till, drainage of water out of the till at rate $C_d$ (mass leaving the subglacial hydrologic system, for example, to deep groundwater storage), and overflow to the distributed drainage system, $\gamma$:

$$\frac{\partial W_{till}}{\partial t} = \frac{m_b}{\rho_w} - C_d - \gamma_t. \tag{45}$$

In the model, meltwater (from either the bed or drained from the surface), is first delivered to the till component. Water in excess of the the maximum storage capacity of the till, $W_{till}^{max}$, is instantaneously transferred as a source term to the distributed drainage system through the $\gamma_t$ term.

### 5.1.2 Distributed drainage

The distributed drainage component is implemented as a "macroporous sheet" that represents bulk flow through linked cavities that form in the lee of bedrock bumps as the glacier slides over the bed (Flowers and Clarke, 2002; Hewitt, 2011; Flowers, 2015). Water flow in the system is driven by the gradient of the hydropotential, $\phi$, defined as

$$\phi = \rho_w g z_b + P_w \tag{46}$$

where $P_w$ is the water pressure in the distributed drainage system. A related variable, the ice effective pressure, $N$, is the difference between ice overburden pressure and water pressure in the distributed drainage system, $P_w$:

$$N = \rho g H - P_w. \tag{47}$$

The evolution of the area-averaged cavity space is a balance of opening of cavity space by the glacier sliding over bedrock bumps and closing through creep of the ice above. The model uses the commonly used assumption (e.g. Schoof, 2010; Hewitt, 2011; Werder et al., 2013; Hoffman and Price, 2014) that cavities always remain water filled (c.f. Schoof et al., 2012), so cavity space can be represented by the effective water depth in the macroporous sheet, $W$:

$$\frac{\partial W}{\partial t} = c_s |\boldsymbol{u_b}|(W_r - W) - c_{cd} A_b N^3 W \tag{48}$$

where $c_s$ is bed roughness parameter, $W_r$ is the maximum bed bump height, $c_{cd}$ is creep scaling parameter representing geometric and possibly other effects, and $A_b$ is the ice flow parameter of the basal ice.

Water flow in the distributed drainage system, $\boldsymbol{q}$, is driven the hydropotential gradient and is described by a general power-law:

$$\boldsymbol{q} = -k_q W^{\alpha_1} |\nabla \phi|^{\alpha_2 - 2} \nabla \phi \tag{49}$$

where $k_q$ is a conductivity coefficient. The $\alpha_1$ and $\alpha_2$ exponents can be adjusted so that Eq. (49) reduces to commonly used water flow relations, such as Darcy flow, the Darcy-Weisbach relation, and the Manning equation.

### 5.1.3 Channelized drainage

The inclusion of channelized drainage in MALI is an extension to the model of Bueler and van Pelt (2015). The distributed drainage model ignores dissipative heating within the water, which in the real world leads to melting of the ice roof, and the formation of discrete, efficient channels melted into the ice above when the distributed discharge reaches a critical threshold (Schoof, 2010; Hewitt, 2011; Werder et al., 2013; Flowers, 2015). These channels can rapidly evacuate water from the distributed drainage system and lower water pressure, even under sustained meltwater input (Schoof, 2010; Hewitt, 2011; Werder et al., 2013; Hoffman and Price, 2014; Flowers, 2015).

The implementation of channels follows the channel network models of Werder et al. (2013) and Hewitt (2013). The evolution of channel area, $S$, is a balance of opening and closing processes as in the distributed system, but in channels the opening

mechanism is melting caused by dissipative heating of the ice above:

$$\frac{dS}{dt} = \frac{1}{\rho L}(\Xi - \Pi) - c_{cc}A_b N^3 S \tag{50}$$

where $c_{cc}$ is the creep scaling parameter for channels.

The channel opening rate, the first term in Eq. (50), is itself a balance of dissipation of potential energy, $\Xi$, and sensible heat change of water, $\Pi$, due to changes in the pressure-dependent melt temperature. Dissipation of potential energy includes energy produced by flow in both the channel itself and a small region of the distributed system along the channel:

$$\Xi = \left|\frac{d\phi}{ds}\boldsymbol{Q}\right| + \left|\frac{d\phi}{ds}\boldsymbol{q_c}l_c\right| \tag{51}$$

where $s$ is the spatial coordinate along a channel segment, $\boldsymbol{Q}$ is the flow rate in the channel, and $\boldsymbol{q_c}$ is the flow in the distributed drainage system parallel to the channel within a distance $l_c$ of the channel. The term adding the contribution of dissipative melting within the distributed drainage system near the channel is included to represent some of the energy that has been ignored from that process in the description of the distributed drainage system and allows channels to form even when channel area is initially zero if discharge in the distributed drainage system is sufficient (Werder et al., 2013). The term representing sensible heat change of the water, $\Pi$, is necessitated by the assumption that the water always remains at the pressure-dependent melt temperature of the water. Changes in water pressure must therefore result in melting or freezing:

$$\Pi = -c_t c_w \rho_w \left(\boldsymbol{Q} + l_c \boldsymbol{q_c}\right)\frac{dP_w}{ds} \tag{52}$$

where $c_t$ is the Clapeyron slope and $c_w$ is the specific heat capacity of water. The pressure-dependent melt term can be disabled in the model.

Water flow in channels, $\boldsymbol{Q}$, mirrors Eq. (49):

$$\boldsymbol{Q} = -k_Q S^{\alpha_1}|\nabla\phi|^{\alpha_2 - 2}\nabla\phi \tag{53}$$

where $k_Q$ is a conductivity coefficient for channels.

### 5.1.4 Drainage component coupling

Eqs. (45)-(53) are coupled together by describing the drainage system with two equations, mass conservation and pressure evolution. Mass conservation of the subglacial drainage system is described by

$$\frac{\partial W}{\partial t} + \frac{\partial W_{till}}{\partial t} = -\nabla \cdot (\boldsymbol{V_d}W) + \nabla \cdot (D_d\nabla W) - \left[\frac{\partial S}{\partial t} + \frac{\partial Q}{\partial s}\right]\delta(x_c) + \frac{m_b}{\rho_w} \tag{54}$$

where $V_d$ is water velocity in the distributed flow, $D_d$ is the diffusivity of the distributed flow, and $\delta(x_c)$ is the Dirac delta function applied along the locations of the linear channels.

Combining Eq. (54) and Eq. (48) and making the simplification that cavities remain full at all times yields an equation for water pressure within the distributed drainage system, $P_w$:

$$\frac{\phi_0}{\rho_w g}\frac{\partial P_w}{\partial t} = -\nabla \cdot \boldsymbol{q} + c_s|\boldsymbol{u_b}|(W_r - W) - c_{cd}A_b N^3 W - \left[\frac{\partial S}{\partial t} + \frac{\partial Q}{\partial s}\right]\delta(x_c) + \frac{m_b}{\rho_w} - \frac{\partial W_{till}}{\partial t} \tag{55}$$

where $\phi_0$ is an englacial porosity used to regularize the pressure equation. Following Bueler and van Pelt (2015), the porosity is only included in the pressure equation and is excluded from the mass conservation equation.

Any of the three drainage components (till, distributed drainage, channelized drainage) can be deactivated at runtime. The most common configuration currently used is to run with distributed drainage only.

### 5.1.5 Numerical implementation

The drainage system model is implemented using Finite Volume Methods on the unstructured grid used by MALI. State variables $(W, W_{till}, S, P_w)$ are located at cell centers and velocities and fluxes $(\boldsymbol{q}, \boldsymbol{V_d}, \boldsymbol{Q})$ are calculated at edge midpoints. Channel segments exist along the lines joining neighboring cell centers. Equation (54) is evaluated by summing tendencies from discrete fluxes into or out of each cell. First-order upwinding is used for advection. At land-terminating ice sheet boundaries, $P_w = 0$ is applied as the boundary condition. At marine-terminating ice sheet boundaries, the boundary condition is $P_w = -\rho_w g z_b$, where $\rho_w$ is ocean water density. The drainage model uses explicit forward Euler time-stepping using Eqs. (45), (54), (50), and (55). This requires obeying advective and diffusive Courant–Friedrichs–Lewy (CFL) conditions for distributed drainage as described by Bueler and van Pelt (2015), as well as an additional advective CFL condition for channelized drainage, if it is active.

We acknowledge that the non-continuum implementation of channels can make the solution grid-dependent, and grid convergence may therefore not exist for many problems (Bueler and van Pelt, 2015). However, for realistic problems with irregular bed topography, we have found dominant channel location is controlled by topography, mitigating this issue.

### 5.1.6 Coupling to ice sheet model

The subglacial drainage model is coupled to the ice dynamics model through a basal friction law. Currently, the only option is a modified Weertman-style power law (Bindschadler, 1983; Hewitt, 2013) that adds a term for effective pressure to Eq. (15):

$$\tau_{bi} = C_0 N u_{bi}^m, \ \ i, j = 1, 2 \tag{56}$$

where $C_0$ is a friction parameter. Implementation of a Coulomb friction law (Schoof, 2005; Gagliardini et al., 2007) and a plastic till law (Tulaczyk et al., 2000; Bueler and van Pelt, 2015) are in development. When the drainage and ice dynamics components are run together, coupling of the systems allows the negative feedback described by (Hoffman and Price, 2014) where elevated water pressure increases ice sliding and increased sliding opens additional cavity space, lowering water pressure. The meltwater source term, $m$, is calculated by the thermal solver in MALI. Either or both of the ice dynamics and thermal solvers can be disabled, in which case the relevant coupling fields can be prescribed to the drainage model.

### 5.1.7 Verification and Real-world Application

To verify the implementation of the distributed drainage model, we use the nearly exact solution described by Bueler and van Pelt (2015). The problem configuration uses distributed drainage only on a two-dimensional, radially-symmetric ice sheet of radius 22.5 km with parabolic ice sheet thickness and a nontrivial sliding profile. Bueler and van Pelt (2015) showed that this

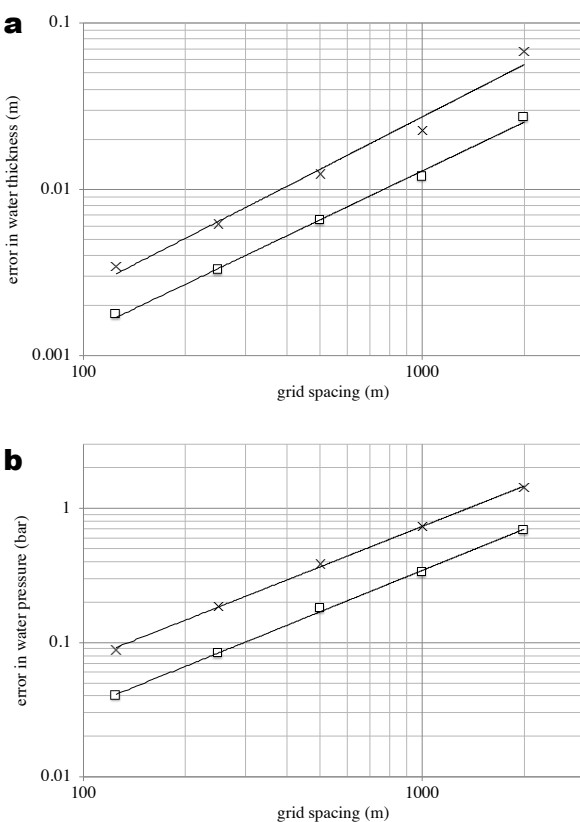

**Figure 3.** Error in subglacial hydrology model for radial test case with near-exact solution described by Bueler and van Pelt (2015) for different grid resolutions. a) Error in water thickness. $\times$ symbols indicate maximum error, and squares indicate mean error. Average error in water thickness decays as $O(\Delta x^{0.97})$. b) Error in water pressure, with same symbols. Average error in water pressure decays as $O(\Delta x^{1.02})$.

configuration allows for nearly-exact reference values of $W$ and $P_w$ to be solved at steady state from an ordinary differential equation initial value problem with very high accuracy. We follow the test protocol of Bueler and van Pelt (2015) and initialize the model with the near-exact solution and then run the model forward for one month, after which we evaluate model error due to drift away from the expected solution. Performing this test with the MALI drainage model, we find error comparable to that found by Bueler and van Pelt (2015) and approximately first order convergence (Fig. 3).

To check the model implementation of channels, we use comparisons to other, more mature drainage models through the Subglacial Hydrology Model Intercomparison Project (SHMIP)[5]. Steady state solutions of the drainage system effective pressure, water fluxes, and channel development for an idealized ice sheet with varying magnitudes of meltwater input (SHMIP experiment suites A and B) compared between MALI and other models of similar complexity (GlaDS, Elmer) are very similar.

To demonstrate a real-world application of the subglacial hydrology model, we perform a standalone subglacial hydrology simulation of the entire Antarctica Ice Sheet on a uniform 20 km resolution mesh (Fig. 4). We force this simulation with basal

---

[5]https://shmip.bitbucket.io/

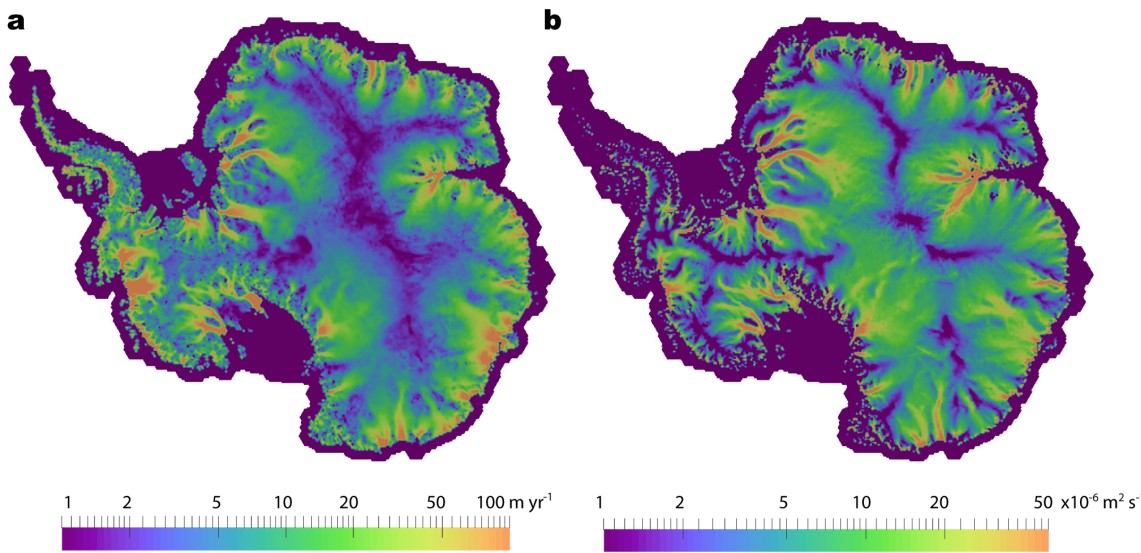

**Figure 4.** Demonstration of subglacial hydrology model capability using 20 km resolution simulation of Antarctica (too coarse resolution for scientific validity but sufficient for demonstrating model capabilities). Subglacial hydrology model results for 20 km resolution Antarctic Ice Sheet. a) Grounded basal ice speed calculated by the first-order velocity solver optimized to surface velocity observations. This field and the calculated basal melt are the forcings applied to the standalone subglacial hydrology model. b) Water flux in the distributed system calculated by the subglacial hydrology model at steady state. (Ice dynamics is prescribed.)

sliding and basal melt rate after optimizing the first-order velocity solver optimized to surface velocity observations (Fig. 4a). We then run the subglacial hydrology model to steady state with only distributed drainage active and using standard parameter values and prescribed ice dynamic forcing. Though this mesh is too coarse to provide scientifically valid results, the modeled subglacial hydrologic state is reasonable. For example, the subglacial water flux increases downglacier and is greatest in fast-flowing outlet glaciers and ice streams (Fig. 4b), as expected from theory and seen in other subglacial hydrology models (e.g., Le Brocq et al., 2009). Calibrating parameters for the subglacial hydrology model and a basal friction law and performing coupled subglacial-hydrology/ice-dynamics simulations are beyond the scope of this paper; we merely mean to demonstrate plausible behavior from the subglacial hydrology model for a realistic ice-sheet-scale problem.

### 5.2 Iceberg Calving

MALI includes a few simple methods for removing ice from calving fronts during each model time step:

1. All floating ice is removed.

2. All floating ice in cells with an ocean bathymetry deeper than a specified threshold is removed.

3. All floating ice thinner than a specified threshold is removed.

4. The calving front is maintained at its initial location by adding or removing ice after thickness evolution is complete. When ice is completely lost in a grid cell through evolution, it is replaced with a thin layer of ice (default value of 1 m). This does *not* conserve mass or energy but provides a simple way to maintain a realistic ice shelf extent (e.g., for model spinup).

5. "Eigencalving" scheme (Levermann et al., 2012). Calving front retreat rate, $C_v$, is proportional to the product of the principal strain rates $(\dot{\epsilon}_1, \dot{\epsilon}_2)$ if they both are extensional:

$$C_v = K_2 \dot{\epsilon}_1 \dot{\epsilon}_2 \text{ for } \dot{\epsilon}_1 > 0 \text{ and } \dot{\epsilon}_2 > 0. \tag{57}$$

The eigencalving scheme can optionally also remove floating ice at the calving front with thickness below a specified thickness threshold (Feldmann and Levermann, 2015). In practice we find this is necessary to prevent formation of tortuous ice tongues and continuous, gradual extension of some ice shelves along the coast.

Ice that is eligible for calving can be removed immediately or fractionally each time step based on a calving timescale. To allow ice shelves to advance as well as retreat, we implement a simple parameterization for sub-grid motion of the calving front by forcing floating cells adjacent to open ocean to remain dynamically inactive until ice thickness there reaches 95% of the minimum thickness of all floating neighbors. This is an *ad hoc* alternative to methods tracking the calving front position at sub-grid scales (Albrecht et al., 2011; Bondzio et al., 2016). In Sect. 8 below, we demonstrate the eigencalving scheme applied to a realistic Antarctic ice sheet simulation. More sophisticated calving schemes are currently under development.

To demonstrate a real-world application of the eigencalving parameterization, we perform a 1000 year spinup of Antarctica with evolving velocity, geometry, and temperature and active eigencalving (Fig. 5). For the purposes of this demonstration, we use the same uniform, 20 km resolution mesh used in Fig. 4, which is too coarse to accurately resolve grounding line dynamics (see Sect. sect:MISMIP3d), and therefore should not be interpreted as a scientifically realistic simulation. We use an initial internal ice temperature field from Van Liefferinge and Pattyn (2013), and the optimization capability described below in Sect. 6.1 (note that we optimize both $\beta$ and $\gamma$ in this case), along with observed surface velocities from Rignot et al. (2011) to obtain a realistic model initial state (Fig. 5a). For the 1000 year spinup, we apply steady forcing of present-day estimates for surface mass balance and submarine melting (Lenaerts et al. (2012) and Rignot et al. (2013), respectively). For temperature boundary conditions, we apply the steady geothermal flux field from Shapiro and Ritzwoller (2004) and the surface (2 meter) air temperature field from Lenaerts et al. (2012). We apply eigencalving calving with the $K_2$ parameter tuned individually for large ice shelves and a minimum calving front thickness threshold of 100 m. This spin-up, albeit much too short to come to full to equilibrium, allows ample time for migration of the calving front and grounding line and removes a substantial portion of the largest model transients. It demonstrates that a tuned eigencalving parameterization is capable of maintaining stable and realistic calving front positions in MALI during ice sheet evolution (Fig. 5a,b), consistent with its implementation in other models (Levermann et al., 2012; Feldmann and Levermann, 2015).

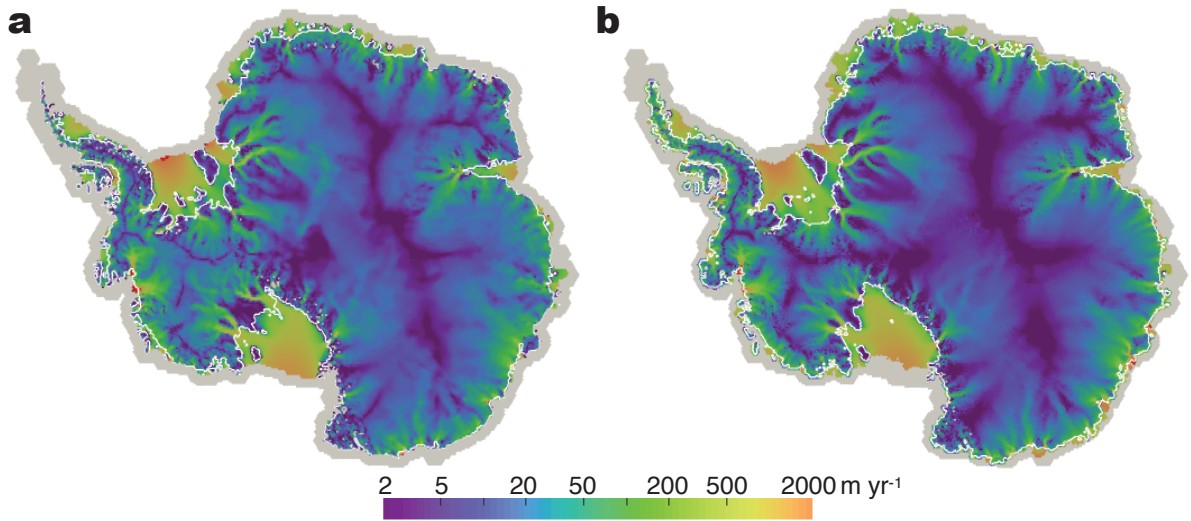

**Figure 5.** Demonstration of eigencalving capability using using 20 km resolution simulation of Antarctica (too coarse resolution for scientific validity but sufficient for demonstrating model capabilities). a) Modeled ice extent and surface speed after optimization. White contour line in each plot is the grounding line. Areas colored red exceed the maximum speed shown in the colorbar. Gray areas are ice-free regions of the computational domain. b) Modeled ice extent and surface speed after 1000 years with evolving velocity, geometry, and temperature and active eigencalving, plotted as in a).

## 6 Additional Capabilities

### 6.1 Optimization

MALI includes an optimization capability through its coupling to the Albany-LI momentum-balance solver described in Sect. 4.2.1. We provide a brief overview of this capability here, while referring to Perego et al. (2014) for a complete description of the governing equations, solution methods, and example applications. In general, our approaches are similar to those reported on for other advanced ice sheet modeling frameworks already described in the literature (e.g., Goldberg and Sergienko, 2011; Larour et al., 2012; Gagliardini et al., 2013; Brinkerhoff and Johnson, 2013; Cornford et al., 2015) and we focus here primarily on optimizing the model velocity field relative to observed surface velocities. Briefly, we consider the optimization functional

$$\mathcal{J}(\beta,\gamma) = \oint_{\Sigma} \frac{1}{2\sigma_u^2} |\boldsymbol{u}_s - \boldsymbol{u}_s^{\mathrm{obs}}|^2 \, d\boldsymbol{s} + \frac{c_\gamma}{2} \oint_{\Sigma} |\gamma - 1|^2 \, d\boldsymbol{s} + \mathcal{R}_\beta(\beta) + \mathcal{R}_\gamma(\gamma), \tag{58}$$

where the first term on the right-hand side (RHS) is a cost function associated with the misfit between modeled and observed surface velocities, the second term on the RHS is a cost function associated with the ice stiffness factor, $\gamma$ (see Eq. (5)), and the third and fourth terms on the RHS are Tikhonov regularization terms given by

$$\mathcal{R}_\beta(\beta) = \frac{\alpha_\beta}{2} \oint_{\Sigma} |\nabla\beta|^2 \, d\boldsymbol{s}, \ \ \mathcal{R}_\gamma(\gamma) = \frac{\alpha_\gamma}{2} \oint_{\Sigma} |\nabla\gamma|^2 \, d\boldsymbol{s}. \tag{59}$$

$\sigma_u$ is an estimate for the the standard deviation of the uncertainty in the observed ice surface velocities and the parameter $c_\gamma$ controls how far the ice stiffness factor is allowed to stray from unity in order to improve the match to observed surface velocities. The regularization parameters $\alpha_\beta > 0$ and $\alpha_\gamma > 0$ control the tradeoff between a smooth $\beta$ field and one with higher-frequency oscillations (that may capture more spatial detail at the risk of over-fitting the observations). The optimal values of $\alpha_\beta$ and $\alpha_\gamma$ can be chosen through a standard L-curve analysis. The optimization problem is solved using the Limited Memory BFGS method, as implemented in the Trilinos package ROL[6], on the reduced-space problem. The functional gradient is computed using the adjoint method.

An example application of the optimization capability applied to a realistic, whole-ice-sheet problem is given below in Sect. 8. Hoffman et al. (2018) present another application to the assimilation of surface velocity time series in Western Greenland.

We note that our optimization framework has been designed to be significantly more general than implied by Eq. (58). While not applied here, we are able to introduce additional observational-based constraints (e.g., mass balance terms) and optimize additional model variable (e.g., the ice thickness). These are necessary, for example, when targeting model initial conditions that are in quasi-equlibrium with some applied climate forcing. These capabilities are discussed in more detail in Perego et al. (2014).

## 6.2 Simulation Analysis

As with other climate model components built using the MPAS framework, MALI supports the development and application of "analysis members", which allow for a wide range of run-time-generated simulation diagnostics and statistics output at user specified time intervals. Support tools included with the code release allow for the definition of any number or combination of pre-defined "geographic features" – points, lines ("transects"), or areas ("regions") – of interest within an MPAS mesh. Features are defined using the standard GeoJSON format (Butler et al., 2016) and a large existing database of globally defined features is currently supported[7]. Python-based scripts are available for editing GeoJSON feature files, combining or splitting them, and using them to define their coverage within MPAS mesh files. Currently, MALI includes support for standard ice sheet model diagnostics (see Table 2) defined over the global domain (by default) and / or over specific ice sheet drainage basins and ice shelves (or their combination). Support for generating model output at points and along transects will be added in the future (e.g., vertical samples at ice core locations or along ground-penetrating radar profile lines). In Sect. 8 below we demonstrate the analysis capability applied to an idealized simulation of the Antarctica ice sheet.

## 7 Model Verification and Benchmarks

MALI has been verified by a series of configurations that test different components of the code. In some cases analytic solutions are used, but other tests rely on intercomparison with community benchmarks that have been run previously by many different ice sheet models.

---

[6]https://trilinos.org/packages/rol/

[7]https://github.com/MPAS-Dev/geometric_features

**Table 2.** Standard model diagnostics available for an arbitrary number of predefined geographic regions.

| diagnostic | units |
| --- | --- |
| net ice area and volume | $m^2$, $m^3$ |
| net grounded ice area and volume | $m^2$, $m^3$ |
| net floating ice area and volume | $m^2$, $m^3$ |
| net volume above floatation | $m^3$ |
| minimum, maximum, and mean ice thickness | m |
| net surface mass balance | kg yr$^{-1}$ |
| net basal mass balance | kg yr$^{-1}$ |
| net basal mass balance for floating ice | kg yr$^{-1}$ |
| net basal mass balance for grounded ice | kg yr$^{-1}$ |
| average surface mass balance | m yr$^{-1}$ |
| average basal mass balance for grounded ice | m yr$^{-1}$ |
| average basal mass balance for floating ice | m yr$^{-1}$ |
| net flux due to iceberg calving | kg yr$^{-1}$ |
| net flux across grounding lines | kg yr$^{-1}$ |
| maximum surface and basal velocity | m yr$^{-1}$ |

MALI currently includes 86 automated system regression tests that run the model for various problems with analytic solutions or community benchmarks. In addition to checking the accuracy of model answers, some of the tests check that model restarts give bit-for-bit exact answers with longer runs without restart. Some others check that the model gives bit-for-bit exact answers on different numbers of processors. All but 20 of the longer running tests are run every time new features are added to the code, and these tests each also include a check for answer changes. The verification and benchmark descriptions below are the most important examples from the larger test suite.

### 7.1 Halfar analytic solution

In (Halfar, 1981, 1983), Halfar described an analytic solution for the time-evolving geometry of a radially-symmetric, isothermal dome of ice on a flat bed with no accumulation flowing under the shallow ice approximation. This test provides an obvious test of the implementation of the shallow-ice velocity calculation and thickness evolution schemes in numerical ice sheet models and a way to assess model order of convergence (Bueler et al., 2005; Egholm and Nielsen, 2010). Bueler et al. (2005) showed the Halfar test is the zero accumulation member of a family of analytic solutions, but we apply the original Halfar test here.

In our application we use a dome following the analytic profile prescribed by Halfar (1983) with an initial radius of 21213.2 m and an initial height of 707.1 m. We run MALI with the shallow ice velocity solver and isothermal ice for 200 years and then

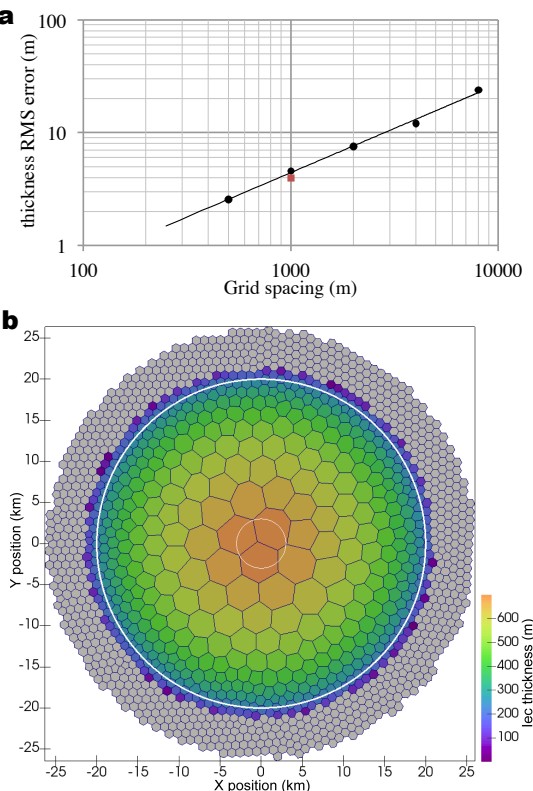

**Figure 6.** a) Root mean square error in ice thickness as a function of grid cell spacing for the Halfar dome after 200 years shown with black dots. The order of convergence is 0.78. The red square show the RMS thickness error for the variable resolution mesh shown in b) with 1000 m spacing around the margin. b) Mesh with resolution that varies linearly from 1000 m grid spacing beyond a radius of 20 km (thick white line) to 5000 m at a radius of 3 km (thin white line). The ice thickness initial condition for the Halfar problem is shown. This mesh requires 1265 cells for the 200 yr duration Halfar test case, while a uniform 1000 m resolution mesh requires 2323 cells.

compare the modeled ice thickness to the analytic solution at 200 years. We find the root mean square error in model thickness decreases as model grid spacing is decreased (Fig. 6a). The order of convergence, 0.78, is somewhat lower than expected from the first-order methods used for advection and time evolution.

We also use this test to assess the accuracy of simulations with variable resolution. We perform an addition run of the Halfar
5   test using a variable resolution mesh that has 1000 m cell spacing beyond a radius of 20 km that transitions to 5000 m cell spacing at a radius of 3 km (Fig. 6b), generated with the JIGSAW(GEO) mesh generation tool (Engwirda, 2017a, b). Root mean square error in thickness for this simulation is similar to that for the uniform 1000 m resolution case (Fig. 6a), providing confidence in the advection scheme applied to variable resolution meshes. The variable resolution mesh has about half the cells of the 1000 m uniform resolution mesh.

## 7.2 EISMINT

European Ice Sheet Modeling Initiative (EISMINT) model intercomparison consisted of two phases designed to provide community benchmarks for shallow-ice models. Both phases included experiments that grow a radially symmetric ice sheet on a flat bed to steady state with a prescribed surface mass balance. The EISMINT intercomparisons test ice geometry evolution and ice temperature evolution with a variety of forcings. Bueler et al. (2007) describe an alternative tool for testing thermo-mechanical shallow-ice models with artificially constructed exact solutions. While their approach has the notable advantage of providing exact solutions, we have not implemented the non-physical three-dimensional compensatory heat source necessary for its implementation. While we hope to use the verification of Bueler et al. (2007) in the future, for now we use the EISMINT intercomparison suites to test our implementation of thermal evolution and thermomechanical coupling.

The first phase (Huybrechts et al., 1996) (sometimes called EISMINT1) prescribes evolving ice geometry and temperature, but the flow rate parameter $A$ is set to a prescribed value so there is no thermomechanical coupling. We have conducted the Moving Margin experiment with steady surface mass balance and surface temperature forcing. Following the specifications described by Huybrechts et al. (1996), we run the ice sheet to steady state over 200 ka. We use the grid spacing prescribed by Huybrechts et al. (1996) (50 km), but due to the uniform Voronoi grid of hexagons we employ, we have a slightly larger number of grid cells in our mesh (1080 vs 961). At the end of the simulation, the modeled ice thickness at the center of the dome by MALI is 2976.7 m, compared with a mean of 2978.0 $\pm$ 19.3 m for the ten three-dimensional models reported by Huybrechts et al. (1996). MALI achieves similar good agreement for basal homologous temperature at the center of the dome with a value of $-13.09°$ C, compared with $-13.34 \pm 0.56°$ C for the six models that reported temperature in Huybrechts et al. (1996).

The second phase of EISMINT (Payne et al., 2000) (sometimes called EISMINT2), uses the basic configuration of the EISMINT1 Moving Margin experiment but activates thermomechanical coupling through Eq. (8). Two experiments (A and F) grow an ice sheet to steady state over 200 ka from an initial condition of no ice, but with different air temperature boundary conditions. Additional experiments use the steady state solution from experiment A (the warmer air temperature case) as the initial condition to perturbations in the surface air temperature or surface mass balance forcings (experiments B, C, and D). Because these experiments are thermomechanically coupled, they test model ice dynamics and thickness and temperature evolution, as well as their coupling. There is no analytic solution to these experiments, but ten different models contributed results, yielding a range of behavior against which to compare additional models. Here we present MALI results for the five such experiments that prescribe no basal sliding (experiments A, B, C, D, F). Our tests use the same grid spacing as prescribed by Payne et al. (2000) (25 km), again with a larger number of grid cells in our mesh (4464 vs 3721).

Payne et al. (2000) report results for five basic glaciological quantities calculated by ten different models, which we have summarized here with the corresponding values calculated by MALI (Table 3). All MALI results fall within the range of previously reported values, except for volume change and divide thickness change in experiment C and melt fraction change in experiment D. However, these discrepancies are close to the range of results reported by Payne et al. (2000), and we consider temperature evolution and thermomechanical coupling within MALI to be consistent with community models, particularly given the difference in model grid and thickness evolution scheme.

**Table 3.** EISMINT2 results for MALI shallow-ice model. For each experiment, model name "EISMINT2" refers to mean and range of models reported in Payne et al. (2000), where we assume the range reported in by Payne et al. (2000) is symmetric about the mean. For experiments B, C, and D reported values are the change from experiment A results. MALI results that lie outside the range of values in Payne et al. (2000) are italicized.

| Experiment | Model | Volume $10^6$ km$^3$ | Area $10^6$ km$^2$ | Melt fraction | Divide thickness m | Divide basal temperature K |
|---|---|---|---|---|---|---|
| A | EISMINT2 | 2.128±0.0725 | 1.034±0.043 | 0.718±0.145 | 3688.3±48.3 | 255.6±1.4 |
|  | MALI | 2.097 | 1.030 | 0.637 | 3671.8 | 255.2 |
| Experiment | Model | % change | % change | % change | % change | change (K) |
| B | EISMINT2 | -2.589±0.4735 | 0.0±0.0 | 11.836±9.3345 | -4.9±0.658 | 4.6±0.259 |
|  | MALI | -2.258 | 0.0 | 16.832 | -5.013 | 4.6 |
| C | EISMINT2 | -28.505±0.602 | -19.515±1.777 | -27.806±15.6855 | -12.9±0.7505 | 3.7±0.3075 |
|  | MALI | *-27.529* | -20.179 | -35.521 | *-12.049* | 3.8 |
| D | EISMINT2 | -12.085±0.618 | -9.489±1.63 | -1.613±2.8725 | -2.2±0.266 | -0.2±0.03 |
|  | MALI | -12.265 | -9.459 | *-5.216* | -2.092 | -0.2 |

A long-studied feature of the EISMINT2 intercomparison is the cold "spokes" that appear in the basal temperature field of all models in Experiment F and, for some models, experiment A (Payne et al., 2000; Saito et al., 2006; Bueler et al., 2007; Brinkerhoff and Johnson, 2013). MALI with shallow-ice velocity exhibits cold spokes for experiment F but not experiment A (Fig. 7). Bueler et al. (2007) argue these spokes are a numerical instability that develops when the derivative of the strain heating term is large. Brinkerhoff and Johnson (2013) demonstrate that the model VarGlaS avoids the formation of these cold spokes. However, that model differs from previously analyzed models in several ways: it solves a three-dimensional, advective-diffusive description of an enthalpy formulation for energy conservation; it uses the Finite Element Method on unstructured meshes; conservation of momentum and energy are iterated on until they are consistent (rather than lagging energy and momentum solutions as in most other models). At present, it is unclear which combination of those features is responsible for preventing the formation of the cold spokes.

## 7.3 Enthalpy benchmarks

Kleiner et al. (2015) present a set of benchmark experiments to test numerical models of ice sheet enthalpy evolution. The experiments use designs that allow comparison to analytic solutions. Here we only give very brief descriptions of the enthalpy benchmark experiments. Details can be found in Kleiner et al. (2015). The benchmark includes two different experiments for testing the capability of transient behavior and horizontal and vertical advection of the enthalpy model. Both experiments use

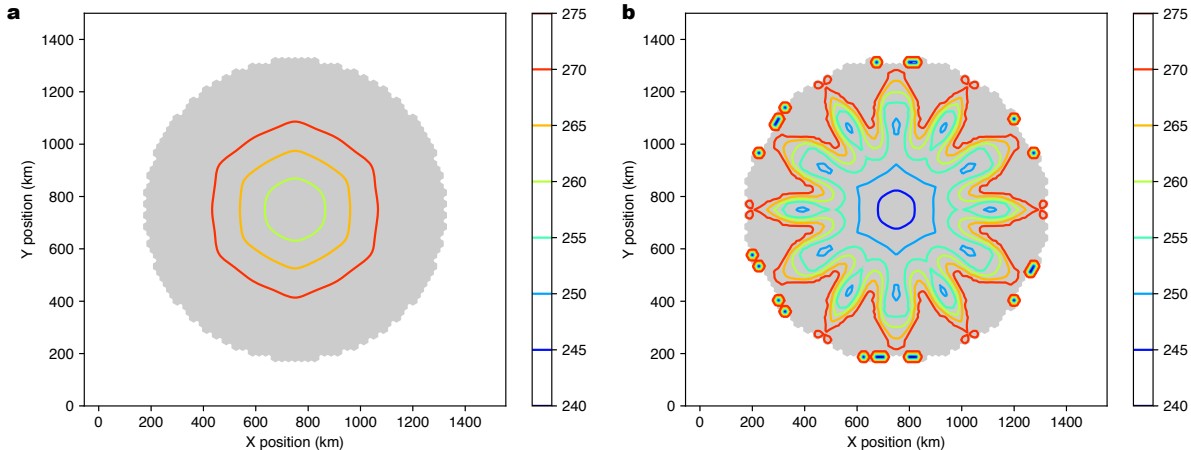

**Figure 7.** a) Basal homologous temperature (K) for EISMINT2 Experiment A. b) Same for Experiment F. Figures are plotted following Payne et al. (2000).

a parallel-sided ice slab with constant thickness and inclination and prescribed ice dynamics decoupled from thermodynamics, resulting in effectively one-dimensional, vertical experiments.

### 7.3.1 Experiment A

Both heat advection and frictional heating are neglected in this experiment. Heat diffusion is the only controlling process in the re-distribution of enthalpy, i.e., the enthalpy balance equation simplifies to

$$\frac{\partial E}{\partial t} = \frac{\partial}{\partial z}\left(K\frac{\partial E}{\partial z}\right) \tag{60}$$

To test the transient ability of the enthalpy model, this experiment was designed to run for a time period of 300 ka. During the run, the geothermal heat flux ($G$) is constant over time, but the surface temperature (upper dirichlet boundary condition) changes in 3 different time intervals.

$$T_s = \begin{cases} -30°\text{C}, & 0 \leq t \leq 100ka \\ -5°\text{C}, & 100 \leq t \leq 150ka^8 \\ -30°\text{C}, & 150 \leq t \leq 300ka \end{cases} \tag{61}$$

During the time period of 100 – 150 ka, when the surface temperature rises to $-5°$ C, the glacier base becomes temperate and the basal boundary condition changes from Neumann-type ($\partial T/\partial z = G$) to Dirichlet-type ($T = T_{pmp}$). Then the surface

---

[8] $T_s = -5°$ C from pers. comm. T. Kleiner compared to $T = -10°$ C incorrectly stated in Kleiner et al. (2015)

temperature switches back to its initial value, $-30°$ C, for testing the reversibility of the enthalpy model. In this experiment the basal water content produced by basal melting is allowed to freely accumulate in order to test the basal melt rate calculation.

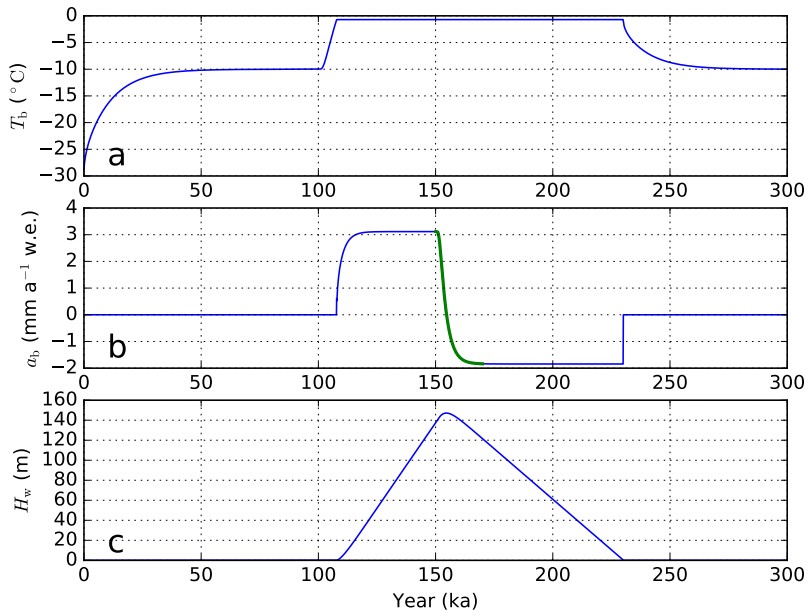

**Figure 8.** The result of a) basal temperature ($T_b$), b) basal melt rate ($a_b$), and c) basal water thickness ($H_w$) for enthalpy benchmark experiment A. The green line from 150 ka to 170 ka in (b) indicates the analytical results (overlapped with model results).

From Fig. 8a, we can clearly see that the basal temperature becomes steady ($-10°$ C) at around 50 ka, and then starts to rise to the pressure melting point after the prescribed change in surface temperature to $-5°$ C at 100 ka. The model then keeps
the glacier base temperate until 225 ka, the prescribed change in the surface temperature boundary condition back to $-30°$ C. The subglacial water layer starts to accumulate from around 110 ka when the basal ice starts to melt (Fig. 8b) and reaches a maximum layer thickness of around 133 m at around 155 ka (Fig. 8c). After the surface gets colder again, it gradually decreases and disappears completely at 225 ka. From the comparison with the analytical basal melt rate result during $150-170$ ka (green line), we can clearly see that the enthalpy model of MALI captures the features of basal melting (water content production).

**7.3.2  Experiment B**

In experiment B, a 200 m thick, $4°$ downward inclined slab is used as the model domain. A particular objective of this experiment is to test the model ability of finding the correct position of the cold-temperate ice transition interface. The horizontal velocity is given as an analytical (shallow-ice approximation type) expression, and the vertical velocity is set to be constant (i.e., thermo-mechanically decoupled). In addition, the geothermal heat flux is set to be zero during the model run so that the
englacial strain heating is the only energy source in the enthalpy balance equation. Initialized from an isothermal field ($-1.5°$

C), our model spins up for several thousand years with a constant surface temperature of $-3°$ C until a steady-state temperate ice layer thickness is achieved (Figure 9).

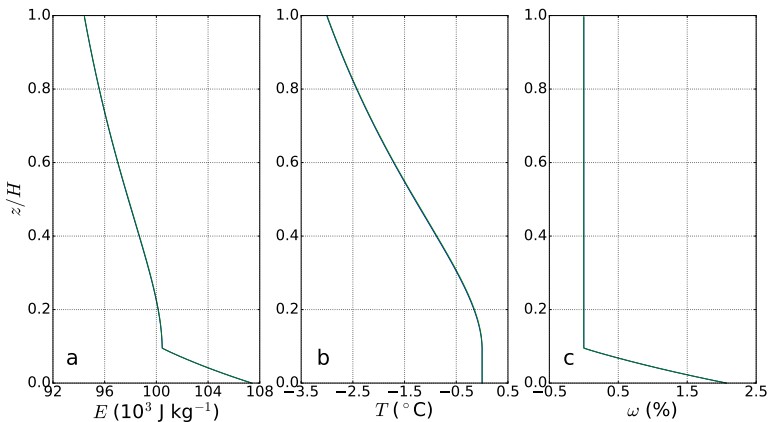

**Figure 9.** The vertical distribution of a) enthalpy ($E$), b) temperature ($T$), and c) water content ($\omega$) for enthalpy benchmark experiment B. The green lines indicate the analytical results (overlapped with model results)

From Figure 2 we can see that our enthalpy model can predict very close enthalpy, temperature and basal water content results (blue lines), compared to analytical solutions (green lines; almost overlapped). Using a uniform vertical resolution
of 1 m, MALI simulates a temperate ice layer thickness of 19 m, nearly identical to the analytical output. This experiment also shows that MALI can accurately compute the englacial enthalpy distribution in the presences of non-zero horizontal ice advection.

### 7.4 ISMIP-HOM

The Ice Sheet Model Intercomparison Project-Higher Order Models (ISMIP-HOM) is a set of community benchmark exper-
10 iments for testing higher-order approximations of ice dynamics (Pattyn et al., 2008). Tezaur et al. (2015a) describe results from the Albany-LI velocity solver for ISMIP-HOM experiments A (flow over a bumpy bed) and C (ice stream flow). For all configurations of both tests, Albany-LI results were within one standard deviation of the mean of first-order models presented in Pattyn et al. (2008), and showed excellent agreement with the similar first-order model formulation of Perego et al. (2012). These tests only require a single diagnostic solve of velocity, and thus results through MALI match those of the standalone
Albany-LI code it is using.

### 7.5 MISMIP3d

The Marine Ice Sheet Model Intercomparison Project-3d (MISMIP3d) is a community benchmark experiment testing grounding line migration of marine ice sheets and includes nontrivial effects in all three dimensions (Pattyn et al., 2013). The exper-

iments use a rectangular domain that is 800 km long in the longitudinal direction and 50 km wide in the transverse direction, with the transverse direction making up half of a symmetric glacier. The bedrock forms a sloping plane below sea level. The first phase of the experiment (Stnd) is to build a steady state ice sheet from a spatially uniform positive surface mass balance, with a prescribed flow rate factor $A$ (no temperature calculation or coupling) and prescribed basal friction for a nonlinear basal

friction law. A marine ice sheet forms with an unbuttressed floating ice shelf that terminates at a fixed ice front at the edge of the domain. From this steady state, the P75R perturbation experiment reduces basal friction by a maximum of 75% across a Gaussian ellipse centered where the Stnd grounding line position crosses the symmetry axis. The perturbation is applied for 100 years, resulting in a curved grounding line that is advanced along the symmetry axis. After the completion of P75S, a reversibility experiment named P75R removes the basal friction perturbation and allows the ice sheet to relax back towards the

Stnd state.

Pattyn et al. (2013) report results from 33 models of varying complexity and applied at resolution ranging from 0.1 to 20 km. Participating models used depth-integrated shallow-shelf or L1L1/L2L2 approximations, hybrid shallow-ice/shallow-shelf approximation, or the complete Stokes equations; there were no three-dimensional first-order approximation models included. This relatively simple experiment revealed a number of key features necessary to accurately model even a simple marine ice

sheet. Insufficient grid resolution prevented reversibility of the steady state grounding line position after experiments P75S and P75R. Reversibility required grid resolution well below 1 km without a subgrid parameterization of grounding line position, and grids a couple times coarser with a grounding line parameterization (Pattyn et al., 2013; Gladstone et al., 2010). The steady state grounding line position in the Stnd experiment was dependent on the stress approximation employed, with Stokes model calculating grounding lines the farthest upstream and models that simplify or eliminate vertical shearing (e.g., shallow shelf)

having grounding lines farther downstream, by up to 100 km. With these features resolved, numerical error due to grounding line motion is smaller than errors due to parameter uncertainty (Pattyn et al., 2013).

We find MALI using the Albany-LI first-order velocity solver is able to resolve the MISMIP3d experiments satisfactorily compared to the Pattyn et al. (2013) benchmark results when using a grid resolution of 500 m with grounding line parameterization. Results at 1 km resolution with grounding line parameterization are close to fully resolved. We first assess grid

convergence by comparing the position of the steady state grounding line in the Stnd experiment for a range of resolutions against our highest resolution configuration of 250 m (Fig. 10). With the grounding line parameterization, the grounding line positions at 500 m and 250 m resolution are very similar (differing by less than the grid resolution), whereas without the grounding line parameterization the grounding line positions in our two highest resolution simulations still differ by 6 km. The converged grounding line position for the Stnd experiment with MALI is 533 km. The grounding line position from our

three-dimensional, first-order stress approximation model falls between that of the L1L2 and Stokes models reported by Pattyn et al. (2013), consistent with the intermediate level of approximation of our model. The dissertation work by Leguy (2015) reported similar results for the Blatter-Pattyn velocity solver in the Community Ice Sheet Model (Lipscomb et al., 2018) when using a subgrid grounding line parameterization.

Reversibility of the P75S and P75R experiments shows the same grid resolution requirement of 500 m, while the 1 km

simulation with grounding line parameterization is close to reversible (Fig. 11). Our highest resolution 250 m simulation

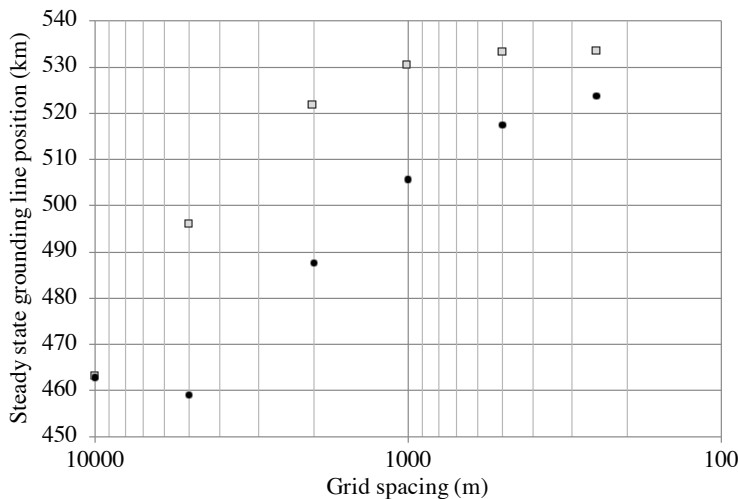

**Figure 10.** Grid resolution convergence for MISMIP3d Stnd experiment with (gray squares) and without (black circles) grounding line parameterization.

without grounding line parameterization does show reversibility at the end of P75R (not shown), but the results differ somewhat from the runs with grounding line parameterization due to the differing starting position determined from the Stnd experiment. Thus for marine ice sheets with similar configuration to the MISMIP3d test, we recommend using MALI with the grounding line parameterization and a resolution of 1 km or less.

The transient results using the MALI three-dimensional first-order stress balance look most similar to those of the "SCO6" L1L2 model presented by Pattyn et al. (2013) in that it takes about 50 years for the grounding line to reach its most advanced position during P75S. In contrast, the Stokes models took notably longer and the models with reduced or missing representation of membrane stresses reached their furthest advance within the first couple decades (Pattyn et al., 2013).

    In addition to MISMIP3d, we have used MALI to perform the MISMIP+ experiments (Asay-Davis et al., 2016). These
results are included in the MISMIP+ results paper in preparation and not shown here.

## 8   Realistic Application: Antarctic Ice Sheet Perturbation Experiment

To demonstrate a large scale, semi-realistic ice sheet simulation that exercises many of the model capabilities discussed above, we describe MALI results from the initMIP-Antarctica experiments. The overall goal of initMIP is to explore the impact of different ice sheet model initialization approaches on simulated ice sheet evolution. Following model initialization – via spin-
up, optimization approaches, or both – three 100 year forward model experiments are conducted in order to examine model response to (1) an unforced control run, (2) an idealized surface mass balance perturbation, and (3) an idealized sub-ice shelf melt perturbation. Here we show results from simulations 1 and 3 for brevity. Additional details of the experiments are described

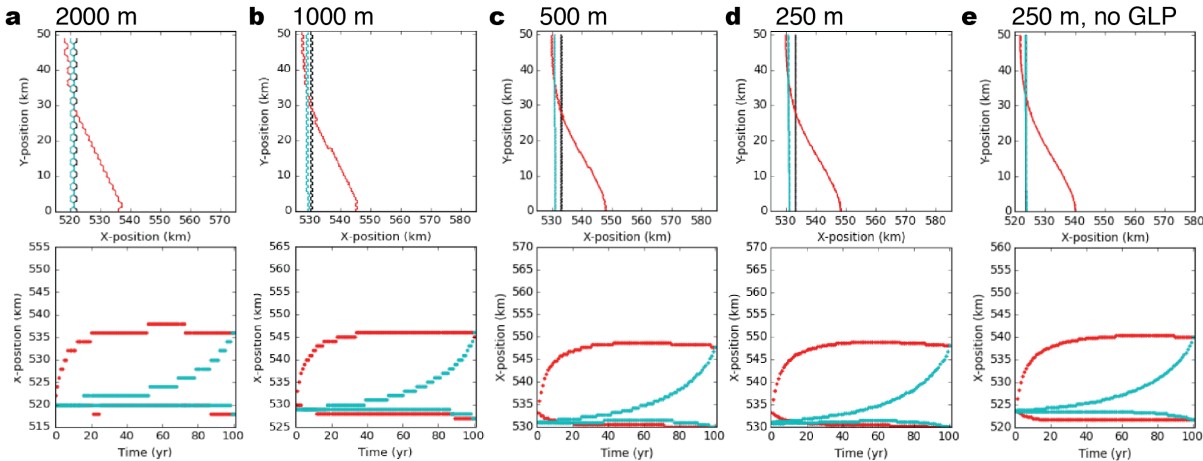

**Figure 11.** Results of the MISMIP3d P75R and P75S experiments from MALI at increasing grid resolution: a) 2000 m, b) 1000 m, c) 500 m, d) 250 m. Results for 250 m without grounding line parameterization (e) are also shown for reference. Plots follow conventions of Figures 5 and 6 in Pattyn et al. (2013). Upper plot in each subplot shows steady state grounding line positions for steady-state spin-up (black), P75S (red), and P75R (blue) experiments. Lower plot in each subplot shows grounding line position with time for P75R (red) and P75S (blue) at y=0 km (top curves) and y=50 km (bottom curves). 500 m and 250 m results are nearly identical. 250 m results without grounding line parameterization are intermediate of those at 1000 m and 2000 m resolution with grounding line parameterization.

in Seroussi and others (2018)[9] and additional details on the broader Ice Sheet Model Intercomparison Project (ISMIP6, part of CMIP6) that initMIP is a part of are described in Nowicki et al. (2016) and Goelzer and others (2018). Additional realistic applications of earlier versions of MALI to Greenland simulations are discussed in Shannon et al. (2013) and Edwards et al. (2014b).

The Antarctica model configuration we use here has 2 km resolution near grounding lines and in regions of marine (below sea level) bedrock in West Antarctica and regions of East Antarctica where present day ice thickness is less than 2500 m to ensure that the grounding line remains in the fine resolution region even under full retreat of West Antarctica and large parts of East Antarctica. The resolution then slowly coarsens to 20 km elsewhere, but maintaining no greater than 6 km resolution on ice shelves (Fig. 12a). The 2 km resolution in regions of potential grounding line migration was chosen as a balance between

acceptable accuracy (Sect. 7.5) and computational cost. The mesh has 1,642,490 horizontal grid cells and uses 10 vertical layers, which are finest near the bed (4% of total thickness) and coarsen towards the surface (23% of total thickness).

    Basal friction ($\beta$) and ice stiffness factor ($\gamma$) fields are optimized as described above in Sect. 6.1 to best allow the model to match observed surface velocities from Rignot et al. (2011). Calving position is fixed as described in option 2 in Sect. 5.2. To avoid energy conservation issues related to maintaining a fixed calving front position, we use a steady internal ice temperature

field from Van Liefferinge and Pattyn (2013). Over the short timescales investigated here (200 yr), we expect the assumption of fixed ice temperature to be a minor uncertainty (Seroussi et al., 2013). From this initial state, we perform a 99 yr relaxation

---

[9]http://www.climate-cryosphere.org/wiki/index.php?title=InitMIP-Antarctica

with evolving velocity and geometry (Fig. 12), applying steady forcing of present-day estimates for surface mass balance and submarine melting (Lenaerts et al. (2012) and Rignot et al. (2013), respectively). This relaxation removes a substantial portion of the largest model transients (Fig. 12c,d). The model state at this point serves as our initial condition from which we run the control and sub-ice shelf melt perturbation experiments mentioned above. In this initial state, the volume above floatation mass

loss for the entire Antarctic Ice Sheet is 602 Gt yr$^{-1}$. This is substantially larger than the current best estimate for Antarctic Ice Sheet mass loss of 109 Gt yr$^{-1}$ for the 1992-2017 period, as well as the larger value of 219 Gt yr$^{-1}$ for the 2012-2017 period (IMBIE team, 2018) and results largely from retreat and thinning in the Thwaites Glacier basin during the 99 yrs of relaxation (Fig. 12d).

The control simulation lost mass above floatation equivalent to a 167 mm sea level rise after 100 yr. The sub-ice shelf melt

perturbation experiment yielded an equivalent of 250 mm sea level rise, an 83 mm sea level rise addition beyond the control run. During the control run, the Ross and Filchner-Ronne ice shelves experienced a modest slowdown with some acceleration near the grounding line (Fig. 13a). Amery and Thwaites ice shelves exhibited significant speedup, with the acceleration propagating inland from Thwaites. Thickness changes in the control run are modest with pronounced thinning occurring only on Thwaites Glacier and inland of Cook Ice Shelf (Fig. 13b). The only noticeable grounding line changes in the control run are a slight

retreat at Thwaites Glacier and a slight advance at Princess Ragnhild Coast in Queen Maud Land. In the sub-ice shelf melt perturbation experiment there is much more pronounced speedup at Thwaites and Pine Island glaciers, propagating far inland (Fig. 13c), with corresponding ice thinning up to 1000 m (Fig. 13d). All other ice shelves show increased thinning or reduced thickening relative to the control run, consistent with the application of increased ice shelf basal melt rates. The regions of Ross and Filchner-Ronne ice shelves near the grounding line experience greater ice speedup than in the control run and some

associated grounding line retreat, but little additional ice thinning. The Totten Glacier region exhibits significant ice speedup and thinning in the sub-ice shelf melt perturbation experiment.

To summarize the disparate regional behavior seen in the experiments, simulation statistics for selected drainage basins for the control and sub-ice shelf melt perturbation experiments experiments are shown in Fig. 14 using the analysis capability described in Sect. 6.2. The applied basal melt forcing for selected basins can be seen in Fig. 14a. The prescribed sub-ice shelf

melt perturbation according to the initMIP protocol increases for 40 years and then remains constant. The continued growth in total ice shelf basal melt after year 40 in Fig. 14a reflects increasing ice shelf area as the grounding line retreats while we have forced the ice shelf calving front to remain fixed. Ice shelf thinning from increase basal melt results in increased flux across the grounding line, with the largest increase in the Thwaites-Pine Island catchment where the ice shelf basal melt perturbation was largest (Fig. 14b). In turn, increased flux across the grounding line leads to ice sheet mass loss (Fig. 14c) and grounding line

retreat (Fig. 14d). The largest changes occur in the Thwaites-Pine Island catchment. Mass loss and grounding line retreat occur in all basins for the control run but are greater in the perturbation experiment, as expected. Note that the initMIP experiments use schematic forcing, and results should not be interpreted as realistic ice sheet or sea level projections. Our aim here is to demonstrate that when applied to large scale, whole-ice-sheet simulations on realistic geometries, MALI is robust and evolves reasonably during multi-century-length, free-running simulations.

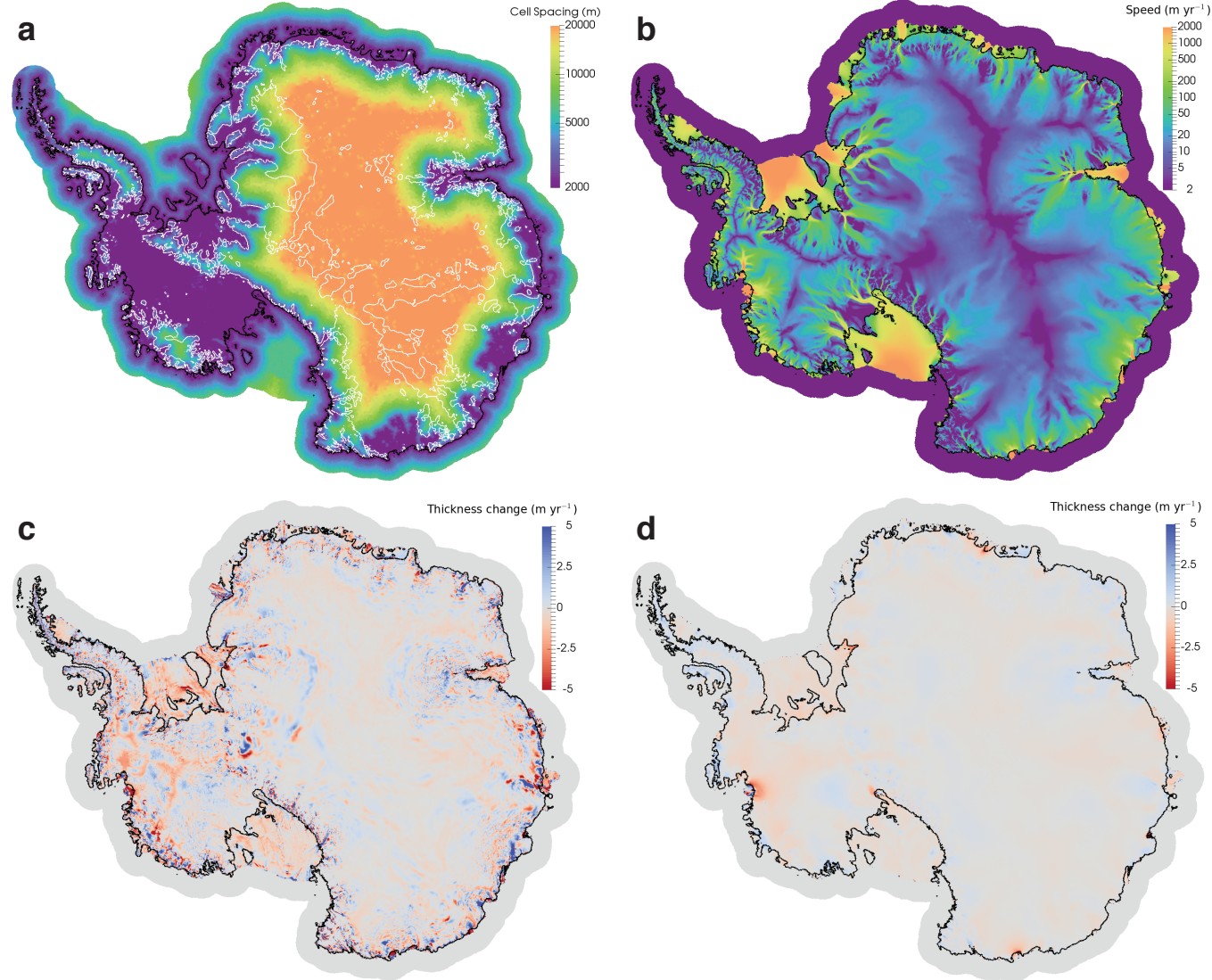

**Figure 12.** (a) Mesh resolution used for initMIP simulations. Black line is the grounding line at the end of the relaxation and white line is the bed topography contour at sea level. (b) Modeled ice surface speed at end of relaxation. Black line is the grounding line. (c) Thickness rate of change at start of relaxation. (d) Thickness rate of change at end of relaxation (99 yr).

## 9    Coupling to Energy Exascale Earth System Model

MALI is the current land ice model component of the U.S. Department of Energy's *Energy Exascale Earth System Model* (E3SM). E3SM is an Earth System Model with atmosphere, land, ocean, and sea ice components, linked through a coupler that passes the necessary fields (e.g., model state, mass, momentum, and energy fluxes) between the components. E3SM, which branched from the Community Earth System Model (CESM, version 1.2 beta10) in 2014, targets high resolution global

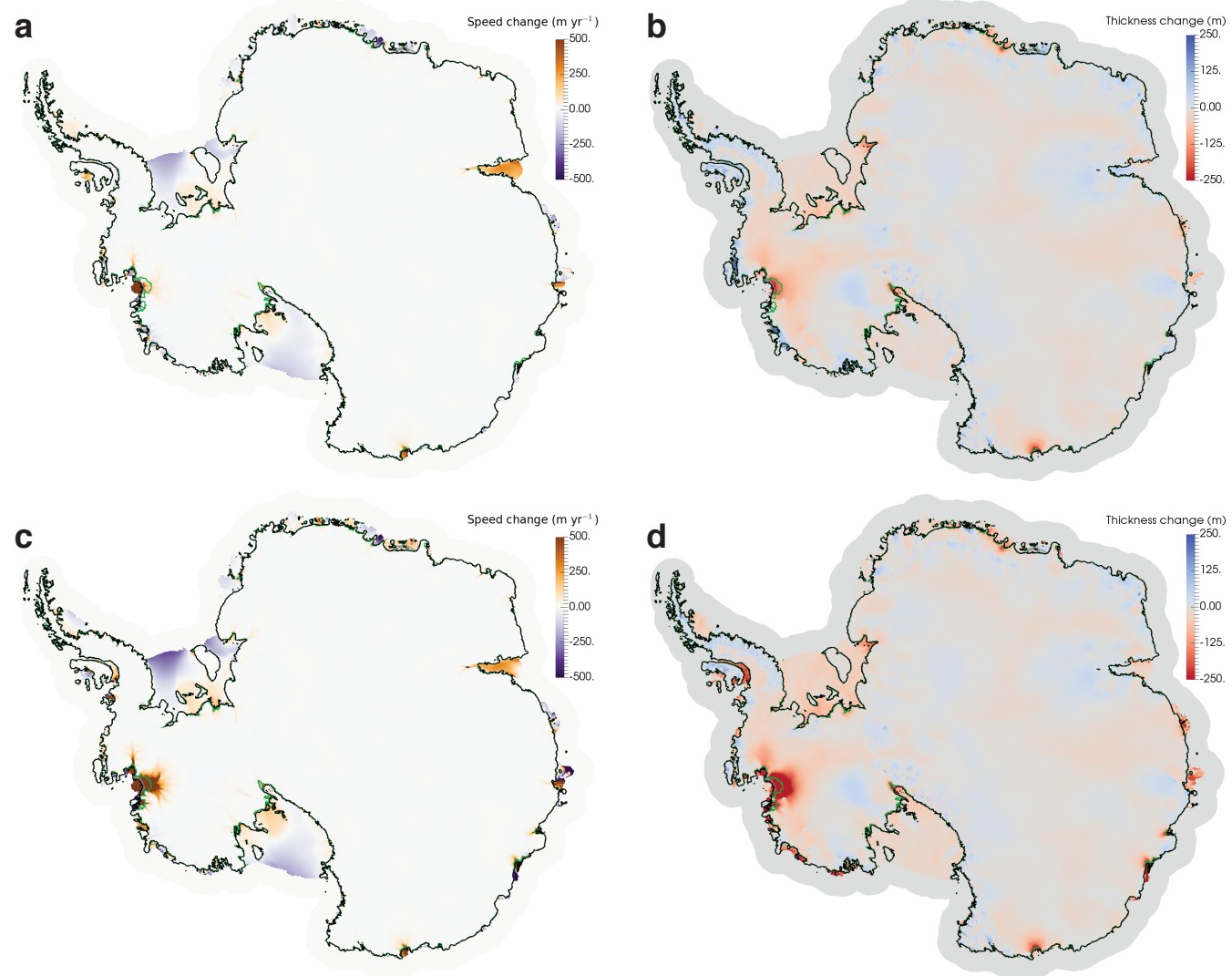

**Figure 13.** Speed and thickness change during 100 yr initMIP simulations. (a) Change in surface speed at 100 yr relative to initial condition for control simulation. (b) Change in ice thickness at 100 yr relative to initial condition for control simulation. (c) Change in surface speed at 100 yr relative to initial condition for sub-ice shelf melt perturbation simulation. (d) Change in ice thickness at 100 yr relative to initial condition for sub-ice shelf melt perturbation simulation. In all panels, the black line indicates the grounding line at the initial time, the gray line is the grounding line at year 100 in the control simulation, and the green line is the grounding line at year 100 in the sub-ice shelf melt perturbation simulation.

simulations, and all components have a variable resolution mesh capability. The ocean (Ringler et al., 2013; Petersen et al., 2015, 2018) and sea ice (Turner et al., 2018) components are also built on the MPAS Framework. Because the coupling between E3SM and MALI is currently still fairly rudimentary, we include only a few additional details below and leave a more detailed

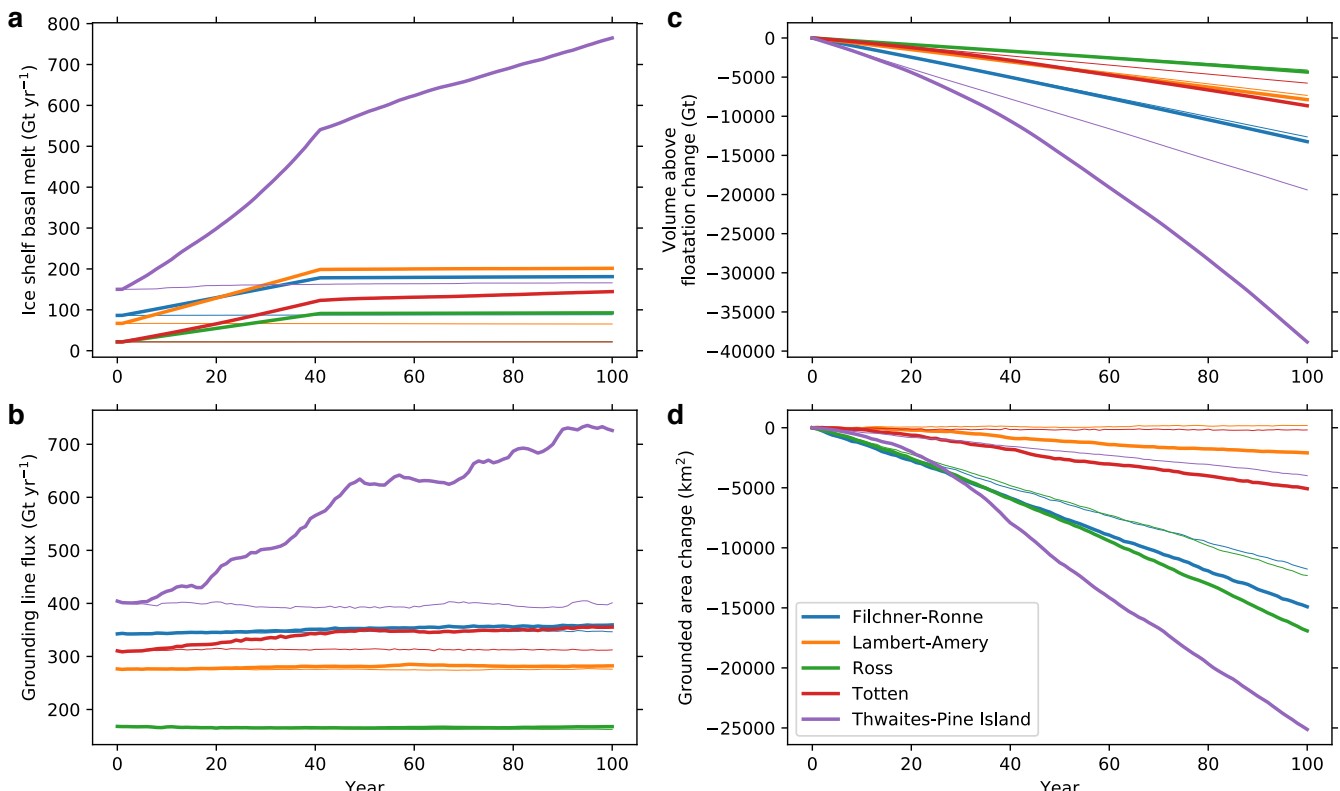

**Figure 14.** Model results for initMIP control (thin lines) and sub-ice shelf melt perturbation (thick lines) experiments for selected drainage basins. Basins are composed of their respective ice shelves and the IMBIE basins (Shepherd et al., 2012) flowing into them from upstream. (a) total ice shelf basal melt; (b) total flux across grounding line; (c) change in volume above floatation from initial time; (d) change in grounded area from initial time.

description to future work. Having all three of these E3SM components in the MPAS framework has simplified adding and maintaining them within E3SM, because developments in the component driver code and build and configuration scripts made by one MPAS component can easily be leveraged by the others. Note that each component of E3SM can be run on differing numbers of processors within the coupled model including the individual MPAS cores.

5    Physics at the ice sheet atmosphere interface are handled by the snow model within the E3SM Land Model (ELM; Zhu et al. (2018); Ricciuto et al. (2018)). ELM's snow model calculates ice sheet surface mass balance using a surface energy balance model and, at each coupling interval, MALI passes the current ice sheet extent and surface elevation through the coupler to ELM. The coupler then returns the surface mass balance and surface temperature calculated by ELM to MALI. These fields are used within MALI as boundary conditions to the mass and thermal evolution equations (Sects. 4.3 and 4.4). Currently, runoff

10   from surface melting is calculated within ELM and routed directly through E3SM's runoff model, rather than being passed to and used by MALI. The subglacial discharge model discussed above in Sect. 5.1 is not currently coupled to the rest of E3SM.

Ongoing and future work on MALI and E3SM coupling includes: passing subglacial discharge at terrestrial ice margins to the land runoff model in E3SM; passing surface runoff calculated in E3SM to the land ice model (for use as a source term in the subglacial hydrology model); two-way coupling between the ocean and a dynamic MALI model[10]; discharge of icebergs (solid ice flux from MALI) to the coupler and from there to the ocean and sea ice models.

## 10  Model Performance

Detailed analysis of the performance and scalability of the Albany/LI velocity solver for idealized test cases and realistic high resolution applications to Greenland and Antarctica has been reported by Tezaur et al. (2015a, b) and Tuminaro et al. (2016). Because the momentum balance solver is $\geq 95\%$ of the cost of a typical forward model time step outside of I/O, the model previously reported performance is generally representative of overall MALI performance. To provide some additional context we summarize MALI performance for the high resolution Antarctica application described in the previous section. That mesh contained 1,642,490 horizontal grid cells and 11 vertical interfaces (10 vertical levels) where the two horizontal components of velocity are solved. The simulations described above were run on the *Edison* Cray XC30 supercomputer[11] at the National Energy Research Scientific Computing (NERSC) Center. Computational nodes on *Edison* each contain two 12-core Intel "Ivy Bridge" processors operating at 2.4 GHz and 64 GB DDR3 1866 MHz memory. Simulations were done using 6400 processors. The control simulation averaged 5.26 simulated years per wall-clock hour (SYPH) over 2031 time steps, and the sub-ice shelf melt perturbation experiment averaged 4.61 SYPH over 2181 time steps. The differing performance is partially due to the longer number of time steps required by the perturbation experiment due to faster maximum ice velocity forcing the adaptive time stepper to take smaller time steps, but may also be a symptom of varying machine performance; performance during different segments of the simulations varied from 3.25–7.38 SYPH presumably due to the usage of different node layouts on the machine and varying I/O performance. On average the velocity solve took 91.9% of the computational time, writing output took 7.5%, and all over operations the remaining 0.6%. Total simulation cost for the 100 year simulations was 122 thousand core-hours for the control run and 139 thousand core-hours for the perturbation simulation. For reference the high resolution E3SM configuration (25 km resolution in atmosphere and land components, varying 18 to 6 km resolution in ocean and sea ice components) runs at 0.12 SYPH using 52,000 processors on *Edison*. While MALI could be run at substantially fewer processors to match the slower throughput of E3SM, the current optimal processor layout for high resolution E3SM could run MALI at the processor count we have done here without incurring any additional expense due to latent processors during model time stepping. At the resolution described here, MALI's computational cost of 1400 processor-hours per simulated year would be insignificant compared to the cost of high resolution E3SM of 448,000 processor-hours per simulated year. Of course running MALI within E3SM would restrict simulation lengths to those used for the coupled model (decades to centuries), which are too short for the investigation of many ice sheet science questions. At E3SM low resolution (100 km resolution in

---

[10]Coupling to a static Antarctic ice sheet with ocean circulation in sub-ice shelf cavities is supported in E3SM version 1.0.0

[11]More information about Edison can be found at http://www.nersc.gov/users/computational-systems/edison/configuration/

atmosphere and land components, varying 60 to 30 km resolution in ocean and sea ice components), the computational cost of MALI within E3SM remains a minor cost.

## 11   Conclusions and Future Work

We have described MPAS-Albany Land Ice (MALI), a higher-order, thermomechanically coupled ice sheet model using unstructured Voronoi meshes. MALI takes advantage of the MPAS framework for development of unstructured grid Earth system model components and the Albany and Trilinos frameworks for parallel, performance-portable, implicit solution of the challenging higher-order ice sheet momentum balance through the Albany-LI velocity solver. Together, these tools provide an accurate, efficient, scalable, and portable ice sheet model targeted for high resolution ice sheet simulations within a larger Earth system modeling framework run on tens of thousands of computing cores, and MALI makes up the ice sheet component of the Energy Exascale Earth System Model version 1.

MALI includes three-dimensional Blatter-Pattyn and shallow-ice velocity solvers, a standard explicit mass evolution scheme, a thermal solver that can use either a temperature or enthalpy formulation, and an adaptive time stepper. Physical processes represented in the model include subglacial hydrology and calving. The model includes a mass-conserving subglacial hydrology model that can represent combinations of water drainage through till, distributed systems, and channelized systems, and can be coupled to ice dynamics. A handful of basic calving schemes are currently implemented, including the physically based "eigencalving" method.

We have demonstrated accuracy of the various model components through commonly used exact solutions and community benchmarks. Of note, we presented the first results for the MISMIP3d benchmark experiments using a Blatter-Pattyn model, and the results are intermediate to those of Stokes and L1L2 models, as might be expected. We also showed simulation results for a semi-realistic Antarctic Ice Sheet configuration at coarse resolution, and this capability was facilitated by the optimization tools within Albany-LI.

A number of model improvements are planned over the next five years, focused heavily on improved representation of ice sheet physical processes and Earth system coupling. An implicit subglacial hydrology model based on existing such models (Werder et al., 2013; Hoffman and Price, 2014) is under development using the Albany framework. It will include optimization capabilities, a technique which has yet to be applied to subglacial hydrology beyond a spatially-average, 0-dimension application (Brinkerhoff et al., 2016). The difficulty in subglacial drainage parameter estimation remains one of the primary reasons drainage models have yet to be widely applied in ice sheet models. Improved calving schemes are also under development using a continuum damage mechanics approach (Borstad et al., 2012; Duddu et al., 2013; Bassis and Ma, 2015). Additionally, solid earth processes affecting ice sheets are planned future development, including gravitational, elastic and viscous effects. Higher-order advection, through the flux-corrected transport (Ringler et al., 2013) and/or incremental remapping (Dukowicz et al., 2010; Lipscomb and Hunke, 2004; Turner et al., 2018) schemes that are already implemented in MPAS, and semi-implicit time-stepping are planned. Finally, a high priority is completing coupling between ice sheet, ocean, and sea ice models in E3SM.

## 12  Code availability

MPAS releases are available at https://mpas-dev.github.io/ and model code is maintained at https://github.com/MPAS-Dev/ MPAS-Release/releases. MPAS-Albany Land Ice is included in MPAS version 6.0. The Digital Object Identifier for MPAS v6.0 is doi:http://doi.org/10.5281/zenodo.1219886. MPAS is openly developed at https://github.com/MPAS-Dev/MPAS-Release.
The Albany library is developed openly at https://github.com/gahansen/Albany, and the Trilinos library is developed openly at https://github.com/trilinos/Trilinos. Region definitions for analysis are openly maintained at https://github.com/MPAS-Dev/ geometric_features.

*Competing interests.*  The authors declare that they have no competing financial interests.

*Disclaimer.*  This paper describes objective technical results and analysis. Any subjective views or opinions that might be expressed in the paper do not necessarily represent the views of the U.S. Department of Energy or the United States Government.
Sandia National Laboratories is a multimission laboratory managed and operated by National Technology and Engineering Solutions of Sandia, LLC., a wholly owned subsidiary of Honeywell International, Inc., for the U.S. Department of Energy's National Nuclear Security Administration under contract DE-NA-0003525.

*Acknowledgements.*  We thank additional contributors to MALI code, Michael Duda, Dominikus Heinzeller, Benjamin Hills, and Adrian Turner, as well as developers of the MPAS, Albany, and Trilinos libraries. Support for this work was provided through the Scientific Discovery through Advanced Computing (SciDAC) program and the Energy Exascale Earth System Model (E3SM) project funded by the US Department of Energy (DOE), Office of Science, Biological and Environmental Research and Advanced Scientific Computing Research and Programs. Development of the subglacial hydrology model was supported by a grant to M.J.H. from the Laboratory Directed Research and Development Early Career Research Program at Los Alamos National Laboratory (20160608ECR). This research used resources of the National Energy Research Scientific Computing Center, a DOE Office of Science User Facility supported by the Office of Science of the U.S. Department of Energy under Contract No. DE-AC02-05CH11231, and resources provided by the Los Alamos National Laboratory Institutional Computing Program, which is supported by the U.S. Department of Energy National Nuclear Security Administration under Contract No. DE-AC52-06NA25396.

**Table A1.** Physical constants.

| Symbol | Description | Standard Value | Units |
|---|---|---|---|
| $g$ | gravitational acceleration | 9.81 | $\text{m s}^{-2}$ |
| $\rho$ | density of ice | 910 | $\text{kg m}^{-3}$ |
| $R$ | gas constant | 8.3145 | $\text{kg m}^2 \text{ s}^{-2} \text{ K}^{-1} \text{ mol}^{-1}$ |
| $\rho_w$ | density of freshwater | 1000 | $\text{kg m}^{-3}$ |
| $\rho_o$ | density of ocean water | 1028 | $\text{kg m}^{-3}$ |
| $c$ | heat capacity of ice | 2009 | $\text{J kg}^{-1} \text{ K}^{-1}$ |
| $k$ | thermal conductivity of ice | 2.1 | $\text{W m}^{-1} \text{ K}^{-1}$ |
| $L$ | latent heat of fusion of water | $3.35 \times 10^5$ | $\text{J kg}^{-1}$ |
| $c_t$ | pressure melt coefficient | $7.5 \times 10^{-8}$ | $\text{K Pa}^{-1}$ |
| $c_w$ | heat capacity of water | $4.22 \times 10^3$ | $\text{J kg}^{-1} \text{ K}^{-1}$ |

**Table A2.** General variables and parameters.

| Symbol | Description | Units |
|---|---|---|
| $x, y \ (x_1, x_2)$ | horizontal coordinates | m |
| $z \ (x_3)$ | vertical elevation | m |
| $nz$ | number of vertical layers | |
| $t$ | time | s |
| $\sigma$ | "sigma" coordinate | unitless |
| $H$ | ice thickness | m |
| $s$ | upper surface elevation | m |
| $b$ | lower surface elevation | m |
| $\dot{a}$ | surface mass balance | $\text{m s}^{-1}$ |
| $\dot{b}$ | basal mass balance | $\text{m s}^{-1}$ |
| $Q_t$ | tracer quantity | |
| $l$ | layer thickness | |
| $\dot{S}$ | tracer sources and sinks | |

**Table A3.** Ice dynamics variables and parameters.

| Symbol | Description | Units |
|--------|-------------|-------|
| $\sigma_{ij}$ | stress tensor | Pa |
| $\tau_{ij}$ | deviatoric stress tensor | Pa |
| $\tau_e$ | effective deviatoric stress | Pa |
| $\delta_{ij}$ | Kroneker delta | |
| $\dot{\epsilon}_{ij}$ | strain rate tensor | $\text{s}^{-1}$ |
| $\dot{\epsilon}_e$ | effective strain rate | $\text{s}^{-1}$ |
| $\mu_e$ | effective ice viscosity | Pa s |
| $\gamma$ | ice stiffening factor | |
| $n$ | Glen's flow law exponent | |
| $m$ | basal friction law exponent | |
| $u_i$ | horizontal ice velocity vector | $\text{m s}^{-1}$ |
| $u_n$ | horizontal advective ice velocity normal to cell edges | $\text{m s}^{-1}$ |
| $\boldsymbol{u_b}$ | basal slip velocity vector | $\text{m s}^{-1}$ |
| $\bar{\mathbf{u}}$ | depth-averaged velocity | $\text{m s}^{-1}$ |
| $A$ | ice flow rate factor | $\text{s}^{-1}\,\text{Pa}^{-n}$ |
| $A_0$ | ice flow rate factor constant | $\text{s}^{-1}\,\text{Pa}^{-n}$ |
| $\beta$ | basal friction coefficient | $\text{Pa yr m}^{-1}$ |

**Table A4.** Ice thermodynamics variables and parameters.

| Symbol | Description | Units |
|--------|-------------|-------|
| $T$ | ice temperature | K |
| $E$ | enthalpy | $\text{J kg}^{-1}$ |
| $T^*$ | absolute ice temperature | K |
| $T_{pmp}$ | pressure melting temperature | K |
| $Q_a$ | activation energy for crystal creep | $\text{kg m}^2\,\text{s}^{-2}\,\text{mol}^{-1}$ |
| $\Phi$ | viscous dissipation | $\text{Pa s}^{-1}$ |
| $F_d$ | diffusive flux at ice base | $\text{W m}^{-2}$ |
| $F_f$ | geothermal heat flux | $\text{W m}^{-2}$ |
| $F_f$ | frictional heating | $\text{W m}^{-2}$ |

**Table A5.** Subglacial hydrology variables and parameters.

| Symbol | Description | Units |
| --- | --- | --- |
| $W_{till}$ | water layer thickness in till | m |
| $W$ | water layer thickness at bed | m |
| $m_b$ | basal melt rate | m s$^{-1}$ |
| $C_d$ | till drainage rate | m s$^{-1}$ |
| $\gamma_t$ | overflow rate from till | m s$^{-1}$ |
| $\phi$ | basal hydropotential | Pa DUPLICATE!!! |
| $P_w$ | basal water pressure | Pa |
| $z_b$ | bed elevation | m |
| $N$ | ice effective pressure | Pa |
| $c_s$ | basal roughness parameter | m$^{-1}$ |
| $W_r$ | maximum bed bump height | m |
| $c_{cd}$ | creep scaling parameter for distributed drainage | |
| $A_b$ | ice flow rate factor for basal ice | s$^{-1}$ Pa$^{-n}$ |
| $\boldsymbol{q}$ | water flow in distributed drainage system | m$^2$ s$^{-1}$ |
| $k_q$ | conductivity coefficient for distributed flow | m$^{2\alpha_2 - \alpha_1}$ s$^{2\alpha_2 - 3}$ kg$^{1-\alpha_2}$ |
| $\alpha_1$ | exponent on water thickness for water flow | |
| $\alpha_2$ | exponent on water pressure for water flow | |
| $S$ | subglacial channel area | m$^2$ |
| $\Xi$ | dissipation of potential energy in water flow | J s$^{-1}$ m$^{-2}$ |
| $\Pi$ | sensible heat change of water | J s$^{-1}$ m$^{-2}$ |
| $c_{cc}$ | creep scaling parameter for channelized drainage | |
| $\boldsymbol{Q}$ | water flow in channelized drainage system | m$^3$ s$^{-1}$ |
| $k_Q$ | conductivity coefficient for channelized flow | m$^{2\alpha_2 - \alpha_1}$ s$^{2\alpha_2 - 3}$ kg$^{1-\alpha_2}$ |
| $\boldsymbol{q_c}$ | water flow in distributed drainage system along a channel | m$^2$ s$^{-1}$ |
| $l_c$ | distance perpendicular to a channel where channel is influenced by distributed flow dissipation | m |
| $V_d$ | water velocity of distributed flow | m s$^{-1}$ |
| $D_d$ | diffusivity of distributed flow | m$^2$ s$^{-1}$ |
| $\delta$ | Dirac delta function | |
| $\phi_0$ | notional englacial porosity | |
| $C_0$ | basal friction parameter | (s m$^{-1}$)$^m$ |

**Table A6.** Calving variables and parameters.

| Symbol | Description | Units |
|---|---|---|
| $\dot{\epsilon}_1, \dot{\epsilon}_2$ | horizontal principal strain rates | $\text{s}^{-1}$ |
| $C_v$ | calving velocity | $\text{m s}^{-1}$ |
| $K_2$ | eigencalving parameter | $\text{m s}$ |

**Table A7.** Optimization variables and parameters.

| Symbol | Description | Units |
|---|---|---|
| $\mathcal{J}$ | optimization functional | |
| $\sigma_u$ | standard deviation of uncertainty in observed velocity | $\text{m s}^{-1}$ |
| $c_\gamma$ | control parameter for ice stiffness factor | |
| $\alpha_\beta, \alpha_\gamma$ | regularization parameters | |

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
