# Peer review of "MPAS-Albany Land Ice (MALI): A variable resolution ice sheet model for Earth system modeling using Voronoi grids"

_Geoscientific Model Development, 2018_

## Short Comment (SC1) · 2 May 2018

Dear authors,

in my role as Executive editor of GMD, I would like to bring to your attention our Editorial version 1.1:

http://www.geosci-model-dev.net/8/3487/2015/gmd-8-3487-2015.html

This highlights some requirements of papers published in GMD, which is also available on the GMD website in the 'Manuscript Types' section:

http://www.geoscientific-model-development.net/submission/manuscript_types.html

In particular, please note that for your paper, the following requirement has not been met in the Discussions paper:

- "The main paper must give the model name and version number (or other unique identifier) in the title."

Please add a version number of MALI in the title of your article in your revised submission to GMD.

Additionally, do not forget to enter the DOI in the code availability section upon final submission!

Yours,

Astrid Kerkweg

---

## Referee Comment (RC1) · S. L. Cornford (Referee) · 15 May 2018

General Comments

This paper is a complete and quite detailed description of the the newly developed MPAS-Albany Land Ice model, which will likely see substantial applications over the next few years, so I expect to see this paper cited often. The model itself makes use of well known process models and robust computational infrastructure, and differs from other models perhaps most in its use of FV discretization on Voronoi polyhedra (cell -centred) for conservation of mass etc, complemented with FEM discretization on Delauney meshes (node-centred).

It seems well suited to GMD, and is well written overall, with a series of tests including convergence studies though I do have a few comments to make.

Specific comments

I suggest removing all section 8. The authors acknowledge that Antarctic ice dynamics are not well described at 20 km resolution in other (conventional) models, and agree that probably applies to MALI too (which is conventional in its treatment), and their own results in earlier sections (especially MISMIP3d) are not distinct from other models in that regard. Yet they claim that the results might be considered 'reasonable'. On what basis? Given that we know that the GL issues are leading order

Don't get me wrong, I appreciate that a realistic example is a good way to show that all the parts (inverse problem, analysis tools etc) work to a reader more interested in ice sheets than models, but how happy would the authors be to have others cite them in future as saying that Antarctic simulations at 20km are reasonable (i.e suggesting that it gives a rough answer, accurate if not precise, rather than an utterly misleading answer).

Is the authors are keen to have a realistic example, and don't have the capacity to demonstrate that MALI actually works well for Antarctica at the required resolution, then perhaps ar regional model might be more suitable.

Technical corrections and minor comments

P2L24. 'adjoint capability' seems like a made up / slang phrase to me. What is needed is the ability to compute gradients / Hessians / Jacobians (depending on what you are trying to achieve) and the ability to solve problems involving the adjoint of a particular operator is a key part of that. I'd say 'differentiation capabilities' if anything.

P6L9. The Jacobian of the residuals of the discrete PDEs, presumably?

P9L16 'The scalability of nonlinear solvers'. Missing definite article? See also L19 re linear solvers.

P9L31 'blue layer interfaces'. Not sure what has happened here, but there seem to be some extra 'blue's scattered about.

P10L21. I think PISM does/did treat SIA explicitly.

L11L11. 'Conservation of mass is used to conduct...' seems awkward. Mass is conserved: ice is transported accordingly.

P12L5. Can you comment on the choice of first order Euler? It's true that other models often end up being first order in time (and space), sometimes without being aware, but few would pick Euler as the first choice.

P12 (all of section 4.4) Aschwanden 2012 described a polythermal energy transport model that is more natural than the cold ice model used here and not much more work to implement. Why cold ice?

P16 Section 5 - 'Additional model physics?'

P22, L19. I'm pretty sure there are no optimization methods in Cornford et al 2013. They are in Cornford 2015.

P23. footote 6 - you *do* have different scalar constants ($\alpha\_\beta$ and $\alpha\_\gamma$) in eq.55. Perhaps you mean something different, but since beta and gamma have such different typical values, you must be scaling the $| grad (X) |^2$ somehow, no?

P25 'The order of convergence *of* 0.78'. of -> is, but more seriuosly, in what sense is 0.78 consistent with 1. 1.02, or 0.98, maybe, but 0.78 ?

L29: L13. Convergence does not as a rule 'occur' when there is an error of $O(h^n)$ so this seems like sloppy language. You mean that, at h < 500m, your results appear converging to be converging at some satisfactory and uniform rate, and are separated by some acceptable tolerance yes?

P29: L17 - you note her that Leguy 2015 had a Blatter-Pattyn model working for this problem, but elsewhere - including the abstract, you claim to be the first.

---

## Referee Comment (RC2) · T Zwinger (Referee) · 21 May 2018

**Review: MPAS-Albany Land Ice (MALI): A variable resolution ice sheet model for Earth system modeling using Voronoi grids, by Hoffman et al.**

**General Impression**

The authors present a new land-ice model, MALI, implemented in the MPAS framework that includes two different solver versions: 1) a directly in the framework implemented lowest order Shallow Ice Solver and 2) a to the framework linked Finite Element (FE) first-order (Blatter-Pattyn) solver based on the code ALBANY. The model is validated against benchmarks, reaching from the EISMINT up to the MISMIP3D tests. Further, the authors present runs of both, the ice sheet as well as the hydrological model on a coarse-mesh Antarctic setup.

The whole project appears to be part of a bigger plan to integrate high-fidelity ice sheet modelling into Earth system models (ESM). This is a positive development and – in particular with the first order approximation – the community will get a tool that should be able to treat marine ice-sheet dynamics to a high accuracy within an ESM framework. Thus, from a general point of view, this publication is a good match to GMD's scope and I in principle support publication.

**Major point of critics**

The Antarctic test cases, both for ice dynamics (spinup) and the hydrology are run on equal-spaced (or equal sized, if talking of tessellations) cells with **horizontal resolution** of 20 km, which in my view is **insufficient to test the physics implemented**. The underlying hydrology as well as the dynamics of the fast outlet systems in Antarctica demand a way higher resolution (as outlet ice-streams usually are of similar or even smaller width). As I see it, one cannot assume that you capture the dynamics of such a system by placing a single cell to resolve it horizontally. In particular, you claim that the hydrology solver is grid-dependent, arguing that usually topography dominates the routing of water, which one certainly not resolves with 20 km. MPAS' and ALBANY's numerics to me seem to be optimized to be deployed on HPC systems as you are utilizing highly parallel scalable linear solvers provided by Trilinos. If so, I suggest to increase the number of cells to overcome that issue. Or – should you by some reason be confined in problem size – just focus on a single outlet system. In that connection I have a further question: On page two, line 21 you mention that your model has the possibility to utilize varying mesh density – why then using so large equal-sized cells in this test case? Could you resolve the critical areas with a sufficient dense mesh?

I like the elegance how you use the connection via the geometric dual between the triangulation of the FE model and the Voronoi tessellation to solve the **transport equations**. That seems to ease the problem of horizontal velocity interpolation. Nevertheless, for me it raises also a question on how **vertical advection** in those equations where it in my view is needed (heat transfer, passive tracer and age/depth) is introduced. You thoroughly explain how you split the vertical diffusion of the heat transfer equation (HTEQ) from the horizontal advection and ignore horizontal diffusion. But: how do you treat vertical advection? Usually, the in SIA and Blatter-Pattyn missing vertical velocity component is obtained from the three-dimensional field of horizontal velocities by utilizing the incompressibility condition, resolving it with respect to the vertical component and integrating it up to a certain z-coordinate (e.g., Greve and Blatter, 2007). Are you performing the same procedure? Else, I would suggest to give a clear explanation why you neglect vertical advection or – in case this applies – how it somehow implicitly is accounted for, for instance within the derivatives of your layer thickness in equation (20). In the latter case, please mention it in the text and

introduce this formulation into the HTEQ, such that the reader can link (22) and (35) to (20). In that connection, it would be from my point of view generally valuable to indicate which vectors and operators are defined in 2D and which in 3D. For instance, I read your HTEQ (22) to be defined for $i=1,2,3$, hence containing a vertical velocity that remains unaccounted for in the description of the numerical implementation given by (35). In my opinion, also the between equations varying index and vector notations do not really help to sort things for the reader.

Thirdly, I am missing an **analysis on the parallel performance** of your implementation. You are mentioning the recent HPC-buzzword "Exascale", but do not really provide any numbers. What would be interesting (at least to me): Is the ice sheet part negligible or significant in terms of computational resources needed if run inside an ESM? How is the performance of the Blatter-Pattyn model compared to a complete Stokes solution and how much more faster is the shallow ice solution compared to the $1^{st}$ order solver? Do you have to adapt a general partitioning scheme for all MPAS-cores, or can you choose the number of CPU-cores to be deployed for each sub-model? Do you provide load balancing in case of changing meshes/domains? Those are questions I think would be valuable information to the reader, as you raise the interest by advertising this code to be massive parallel scalable and to operate within an ESM-framework. If the project has not proceeded so far that you would be able to provide these figures at the moment, then please mention it in the text, as it might be valuable information for people that else might think that this is ready for production.

**Detailed Points**

Points I see to be either corrected or elaborated on, sorted by their occurrence within the text.

**page 3, line 23:** *Shared memory parallelization through OpenMP is also supported, but the implementation is left up to each core.*

What does that mean? I would consider multi-threading to make sense only with multiple cores (talking of CPU-cores) that share a common memory. If you mean "core" in the sense of MPAS model component, then please use "MPAS-cores", as it else might be misleading.

**page 4, line 29:** *Planar meshes can easily be made periodic by taking advantage of the unstructured mesh specification.*

A minor issue, but I cannot follow your argumentation here: Why would the unstructured nature of grids ease the introduction of periodic conditions? In contrary, if I have a nicely structured mesh, the mapping of conformal nodes by indexes should be easier.

**page 4, line 31:** *Each core chooses its own vertical coordinate system.*

Again, I presume this is "core" used in the sense of MPAS model component but by a computational scientist it could be misinterpreted as CPU-core, also in view of a later occurrence (page 6, line 11) where you actually use "core" in the sense of CPU-core: *It is a massively parallel code by design and recently it has been adopting the Kokkos (Edwards et al., 2014a) programming model to provide manycore performance portability (Demeshko et al., 2018) on major HPC platforms*

**page 5, line 18:** *Additionally, the JIGSAW(GEO) mesh generation tool (Engwirda, 2017a, b) can be used to generate high quality variable resolution meshes with data-based density functions very efficiently.*

Please, explain if and how this is relevant to your application. Do you for instance use some Hessian of velocity field in order to manipulate mesh density?

**page 6, line 27. eq. (2):**
$$\frac{\partial \sigma_{ij}}{\partial x_j} + \rho g = 0, \ \ i,j = 1,2,3$$

Gravity should be displayed as a vector in this balance and hence is missing the index $i$ .

**page 7, line 9:** *In Equation 5, $A$ is a temperature dependent rate factor, n is an exponent commonly taken as 3 for polycrystalline glacier ice, and γ is an ice "stiffness" factor (inverse enhancement factor) commonly used to account for other impacts on ice rheology, such as impurities or crystal anisotropy (see also Section 6.1).*

If you are referring to the commonly applied enhancement factor, $E$, as a pre-factor to the rate factor in Glen's flow law, then not its inverse (as at least I would interpret it), $\gamma = E^{-1}$, but rather $\gamma = E^{-1/n}$ would be correct. This might not be unimportant to spell out, as you use this factor in the inversions and – presumably – plug it back into your forward model.

**page 7, line 18:** in which $A_0$ is a constant, $(T^*)$ is the absolute temperature (i.e., corrected for the dependence of melt temperature on ice pressure), $Q_a$ is the activation energy for crystal creep, and $R$ is the gas constant.

I would see $T^*$ (BTW. why do you put it into brackets?)  to be (quoting Greve and Blatter,2007) the *temperature relative to the pressure melting point* and not the *absolute temperature*.

**page 8, line 2:** *Ice sheets typically have a small aspect ratio, small surface and bed slopes, and vertical pressure distributions that are very nearly hydrostatic.*

In the light that you apply Blatter-Pattyn, in my opinion it would be better not to refer to pressure but rather the vertical distribution of the vertical normal Cauchy stress, i.e. the *hydrostatic stress approximation* (Greve and Blatter, 2007).

**page 9, line 3, equation (15):**
$$2\mu\dot{\boldsymbol{\epsilon}}_1 \cdot \mathbf{n} + \beta u^m = 0, \ \ 2\mu\dot{\boldsymbol{\epsilon}}_2 \cdot \mathbf{n} + \beta v^m = 0, \ \ \text{sliding,}$$

If the symbol $\mu$ stands for the effective viscosity (that is what I presume), you are deviating from the symbol $\eta_f$ you used in equation (9) – which by subscript (e -> f) again deviates from its initial definition in equation (5) and also from the definition in the table of symbols - and you should correct it. If it defines something else, then you should declare it. Same accounts for equation (16).  For someone who wants to really understand what you are doing in your code, consistency in notation is very helpful.

**page 9, line 5, equation (16):**
$$2\mu\dot{\boldsymbol{\epsilon}}_i \cdot \mathbf{n} - \rho g(s - z)\mathbf{n} = \rho_o g \max(z,0)\mathbf{n},$$

The first term in equation (16) to me reads a scalar-product between two vectors, which leads to a scalar, but the other terms to me read as a with the surface normal aligned vector. To me this seems to be inconsistent.

**page 12, line 26**: … $k$ *is assumed constant and uniform*

The temperature dependent heat conductivity of ice varies about 34% between atmospheric pressure melting point and -50  C (Greve and Blatter, 2007). As this is a temperature span that easily can occur in a single column of the Antarctic ice sheet, could you please elaborate how big of an error do you introduce by using a constant value, and which value (corresponding to which temperature) you are choosing?

**page 19, line 17, equation (52):**
$$\boldsymbol{\tau_b} = C_0 N \boldsymbol{u_b}^m$$

A minor issue, but this type of notation leaves room for (mis)interpretation of a vector subject to an exponent. Looking at equation (15), I conclude you mean that the exponent applies component-wise. My suggestion: Either mention that in the text or perhaps write (52) out on a component level – just like (15).

**page 21, line 1***: … , suggesting a basal friction law based on the subglacial hydrologic state could be configured to yield realistic ice velocity.*

This links to one of the major points of criticism: I do not think you can draw that conclusion from a run done on 20 km grid spacing, when the hydrology in the major outlet systems takes place on subgrid scale.

**page 21, line 10***: The calving front is maintained at its initial location by adding or removing ice after thickness evolution is complete. This option does not conserve mass or energy but provides a simple way to maintain a realistic ice shelf extent (e.g., for model spinup).*

I don't get the point of this statement: Doesn't this apply to any sort of calving process? In my opinion introduction of discontinuous calving processes in a continuum model, simply by the fact that it instantaneously removes parts of the continuum, by definition drains mass and energy. I guess you referring to the artificially added mass – then it would be interesting to know how you implemented this. Do you add a layer of minimum ice-depth of 1m as on land?

**page 31, line 10***: The Antarctica model configuration we demonstrate here uses a 20 km uniform resolution mesh, …*

As I explained before, I see 20 km resolution as a problem, in particular in connection with the marine ice sheet dynamics. In particular, as you claim yourself on page 29: *Thus for marine ice sheets with similar configuration to the MISMIP3d test, we recommend using MALI with the grounding line parameterization and a resolution of 1 km or less.*

**Typos and type-setting**

**page 7, line 23:** *Ice sheet models solve Eq. 2-8 with …*

Please, check how GMD wants to have references to equations. I see inconsistencies throughout the text, with either the word Equation/s spelled out or abbreviated (like here). Also check, whether you should put the equation numbers into brackets or not.

**page 17, line 7:** *… space can be represented by the effective water depth in the macorporous sheet, W:*

macorporous -> macroporous

**page 35, line 17:** *An implicit subglacial hydrology model based on existing  models (Werder et al., 2013; Hoffman and Price, 2014) is under development using the Albany framework.*

To me this sentence reads strange, except if the word "such" is removed

**page 42, line 7:** Bueler, E., Lingle, C. S., Kallen-Brown, J. a., Covey, D. N., and Bowman, L. N.: Exact solutions and verification of numerical models for isothermal ice sheets, Journal of Glaciology, 51, 291–306, doi:10.3189/172756505781829449, http://openurl.ingenta.com/content/xref?genre=article{&}issn=0022-1430{&}volume=51{&}issue=173{&}spage=291, 2005.

Those longish (and in this case for my browser not existing) links could be dropped - these occur several times throughout the list of references. In general, I think the DOI link is sufficient and no other link is needed. Perhaps GMD has a policy for reference-styles.

**References**

Greve R. and H. Batter (2007): *Dynamics of Ice Sheets and Glaciers*, 1st ed. , Springer, Berlin

---

## Author Comment (AC1) · 17 Aug 2018

**Comments to Reviewers**

Our responses to reviewer comments are in blue font below.

**Executive Editor: A. Kerkweg**

• "The main paper must give the model name and version number (or other unique identifier) in the title."

Please add a version number of MALI in the title of your article in your revised submission to GMD.

We have added the model version number to the title.

Additionally, do not forget to enter the DOI in the code availability section upon final submission!

We have added the final DOI for the code release in the code availability section.

**Reviewer 1: S. L. Cornford**

Specific comments

I suggest removing all section 8. The authors acknowledge that Antarctic ice dynamics are not well described at 20 km resolution in other (conventional) models, and agree that probably applies to MALI too (which is conventional in its treatment), and their own results in earlier sections (especially MISMIP3d) are not distinct from other models in that regard. Yet they claim that the results might be considered 'reasonable'. On what basis? Given that we know that the GL issues are leading order
Don't get me wrong, I appreciate that a realistic example is a good way to show that all the parts (inverse problem, analysis tools etc) work to a reader more interested in ice sheets than models, but how happy would the authors be to have others cite them in future as saying that Antarctic simulations at 20km are reasonable (i.e suggesting that it gives a rough answer, accurate if not precise, rather than an utterly misleading answer). Is the authors are keen to have a realistic example, and don't have the capacity to demonstrate that MALI actually works well for Antarctica at the required resolution, then perhaps a regional model might be more suitable.

We agree with the reviewer of the inappropriateness of 20 km resolution for scientific purposes. While we endeavored to emphasize this in the text, we are sensitive to the hazard of the inclusion of these results as tacit endorsement of unresolved simulations. In our revised manuscript we have replaced the 20 km simulation results with results from a variable resolution grid that maintains 2 km resolution at present day grounding lines and anywhere grounding lines retreat during the simulation. That mesh coarsens to 20 km in the ice sheet interior. The 2km resolution is chosen based on grid convergence analysis

from the MISMIP3d test case, giving us confidence this resolution is scientifically meaningful (even if the schematic forcing of the experiments shown is not). The increased configuration and simulation time of this high resolution mesh is partly to blame for the delayed resubmission of our manuscript.

Technical corrections and minor comments

P2L24. 'adjoint capability' seems like a made up / slang phrase to me. What is needed is the ability to compute gradients / Hessians / Jacobians (depending on what you are trying to achieve) and the ability to solve problems involving the adjoint of a particular operator is a key part of that. I'd say 'differentiation capabilities' if anything.

We replaced "adjoint capability" with "automatic differentiation capability for the computation of adjoint sensitivities".

P6L9. The Jacobian of the residuals of the discrete PDEs, presumably?

Yes, we clarified this in the text.

P9L16 'The scalability of nonlinear solvers'. Missing definite article? See also L19 re linear solvers.

Added definite articles to both phrases.

P9L31 'blue layer interfaces'. Not sure what has happened here, but there seem to be some extra 'blue's scattered about.

Extraneous "blue"s have been removed!

P10L21. I think PISM does/did treat SIA explicitly.

This is a fair comment, but we believe the statement that "SIA models *typically*" solve an implicit problem to be accurate, so we have left the text unchanged. We have chosen not to mention PISM's explicit SIA treatment because PISM's normal mode of operation is to combine SIA with SSA in a hybrid mode that is different from the class of models we are considering here.

L11L11. 'Conservation of mass is used to conduct...' seems awkward. Mass is conserved: ice is transported accordingly.

Arguably, one could choose an ice transport scheme that is not based on conservation of mass. In an attempt to reduce the awkwardness, we have reworded this sentence as "Ice sheet mass transport and evolution is conducted using the principle of conservation of mass."

P12L5. Can you comment on the choice of first order Euler? It's true that other models often end up being first order in time (and space), sometimes without being aware, but

few would pick Euler as the first choice.

First-order Euler was chosen solely for simplicity as choice that is easy to implement in a new model. We have investigated semi-implicit and Runge-Kutta time integration methods, but neither approach is mature enough to include in this paper.

P12 (all of section 4.4) Aschwanden 2012 described a polythermal energy transport model that is more natural than the cold ice model used here and not much more work to implement. Why cold ice?

We have added a description of an enthalpy formulation for the conservation of energy. This was already in the code but had not been verified, so we had left it out of the original manuscript. We have completed the verification of the enthalpy model, so we have added a description of it in addition to the cold ice formulation, and we have added a new section to the "Model Verification and Benchmarks" section showing the enthalpy model verification. The cold ice model description has been retained.

P16 Section 5 - 'Additional model physics?'

Change made.

P22, L19. I'm pretty sure there are no optimization methods in Cornford et al 2013. They are in Cornford 2015.

You would be the definitive person on that topic! Correction made.

P23. footote 6 - you *do* have different scalar constants ($\alpha\_\beta$ and $\alpha\_\gamma$) in eq.55. Perhaps you mean something different, but since beta and gamma have such different typical values, you must be scaling the $|\text{grad}(X)|^2$ somehow, no?

You are correct. The footnote was referring to an older version of the equation and has now been removed.

P25 'The order of convergence *of* 0.78'. of -> is, but more seriuosly, in what sense is 0.78 consistent with 1. 1.02, or 0.98, maybe, but 0.78 ?

This is a fair criticism, and the text has been modified accordingly: "The order of convergence is 0.78, somewhat lower than expected from the first-order methods used for advection and time evolution."

L29: L13. Convergence does not as a rule 'occur' when there is an error of $O(h^n)$ so this seems like sloppy language. You mean that, at h < 500m, your results appear converging to be converging at some satisfactory and uniform rate, and are separated by some acceptable tolerance yes?

We have reworded this to use more precise language:

*"With the grounding line parameterization, the grounding line positions at 500 m and 250 m resolution are very similar (differing by less than the grid resolution), whereas without the grounding line parameterization the grounding line positions in our two highest resolution simulations still differ by 6 km."*

P29: L17 - you note her that Leguy 2015 had a Blatter-Pattyn model working for this problem, but elsewhere - including the abstract, you claim to be the first.

*We were making the distinction of peer-reviewed literature, but to avoid splitting hairs we have removed those assertions.*

**Reviewer 1: Thomas Zwinger**

**Major point of critics**

The Antarctic test cases, both for ice dynamics (spinup) and the hydrology are run on equal-spaced (or equal sized, if talking of tessellations) cells with horizontal resolution of 20 km, which in my view is insufficient to test the physics implemented. The underlying hydrology as well as the dynamics of the fast outlet systems in Antarctica demand a way higher resolution (as outlet ice-streams usually are of similar or even smaller width). As I see it, one cannot assume that you capture the dynamics of such a system by placing a single cell to resolve it horizontally. In particular, you claim that the hydrology solver is grid- dependent, arguing that usually topography dominates the routing of water, which one certainly not resolves with 20 km. MPAS' and ALBANY's numerics to me seem to be optimized to be deployed on HPC systems as you are utilizing highly parallel scalable linear solvers provided by Trilinos. If so, I suggest to increase the number of cells to overcome that issue. Or – should you by some reason be confined in problem size – just focus on a single outlet system. In that connection I have a further question: On page two, line 21 you mention that your model has the possibility to utilize varying mesh density – why then using so large equal-sized cells in this test case? Could you resolve the critical areas with a sufficient dense mesh?

*See response above to the first reviewer. The real-world application section has been replaced with a mesh using 2 km resolution in areas of important ice dynamics. For the demonstration of the subglacial hydrology model and eigencalving parameterization we have retained the coarse 20 km meshes. We do claim to resolve all the important physics at that scale, but merely to demonstrate the functionality of those physics modules. Tuning those physical processes for high resolution has not yet been completed and is beyond the scope of this paper.*

I like the elegance how you use the connection via the geometric dual between the triangulation of the FE model and the Voronoi tessellation to solve the transport equations. That seems to ease the problem of horizontal velocity interpolation. Nevertheless, for me it raises also a question on how vertical advection in those equations where it in my view is needed (heat transfer, passive tracer and age/depth) is introduced. You thoroughly explain how you split the vertical diffusion of the heat transfer equation (HTEQ) from the horizontal advection and ignore horizontal diffusion. But: how do you treat vertical advection? Usually, the in SIA and Blatter-Pattyn missing vertical velocity component is obtained from

the three-dimensional field of horizontal velocities by utilizing the incompressibility condition, resolving it with respect to the vertical component and integrating it up to a certain z-coordinate (e.g., Greve and Blatter, 2007). Are you performing the same procedure? Else, I would suggest to give a clear explanation why you neglect vertical advection or – in case this applies – how it somehow implicitly is accounted for, for instance within the derivatives of your layer thickness in equation (20). In the latter case, please mention it in the text and introduce this formulation into the HTEQ, such that the reader can link (22) and (35) to (20). In that connection, it would be from my point of view generally valuable to indicate which vectors and operators are defined in 2D and which in 3D. For instance, I read your HTEQ (22) to be defined for $i=1,2,3$, hence containing a vertical velocity that remains unaccounted for in the description of the numerical implementation given by (35). In my opinion, also the between equations varying index and vector notations do not really help to sort things for the reader.

The treatment of vertical advection is handled through a vertical remapping step after horizontal advection is complete.  This was described at the end of section 4.3, but it was not explicitly stated that this step serves to produce vertical advection for thickness and tracers.  That paragraph has been modified to more clearly explain how this works, and Section 4.4 has been modified to reiterate how this applies for the advection of temperature as a tracer.

Thirdly, I am missing an analysis on the parallel performance of your implementation. You are mentioning the recent HPC-buzzword "Exascale", but do not really provide any numbers. What would be interesting (at least to me): Is the ice sheet part negligible or significant in terms of computational resources needed if run inside an ESM? How is the performance of the Blatter-Pattyn model compared to a complete Stokes solution and how much more faster is the shallow ice solution compared to the 1st order solver? Do you have to adapt a general partitioning scheme for all MPAS-cores, or can you choose the number of CPU- cores to be deployed for each sub-model? Do you provide load balancing in case of changing meshes/domains? Those are questions I think would be valuable information to the reader, as you raise the interest by advertising this code to be massive parallel scalable and to operate within an ESM-framework. If the project has not proceeded so far that you would be able to provide these figures at the moment, then please mention it in the text, as it might be valuable information for people that else might think that this is ready for production.

We have added a short performance section describing performance for the high-res Antarctica runs that have been added and their relation to the computational cost of E3SM.  Details on performance and scaling of the velocity solver (which makes up the vast majority of the total model cost) have previously been described in 3 publications.  Comparing the cost of MALI to a Stokes model is no easily done without a coordinated comparison exercise with such a model.  Also we do not think a comparison of the Blatter-Pattyn solver in MALI to the SIA solver is meaningful as the SIA solver was written primarily as a prototyping tool and additionally it cannot be run for Antarctica due to the presence of ice shelves.  Each component of E3SM can be run on differing numbers of processors within the coupled model including the individual MPAS cores.  A note in the E3SM section has been added stating this.

**Detailed Points**

Points I see to be either corrected or elaborated on, sorted by their occurrence within the text.

page 3, line 23: Shared memory parallelization through OpenMP is also supported, but the implementation is left up to each core.

What does that mean? I would consider multi-threading to make sense only with multiple cores (talking of CPU-cores) that share a common memory. If you mean "core" in the sense of MPAS model component, then please use "MPAS-cores", as it else might be misleading.

MPAS core was meant here. This has been clarified.

page 4, line 29: Planar meshes can easily be made periodic by taking advantage of the unstructured mesh specification.

A minor issue, but I cannot follow your argumentation here: Why would the unstructured nature of grids ease the introduction of periodic conditions? In contrary, if I have a nicely structured mesh, the mapping of conformal nodes by indexes should be easier.

What is meant is that because of the unstructured nature, there is little difference between specifying relationships between neighboring cells and periodic cells. I agree that finding periodic correspondence between cells is simpler on a structured mesh, but then most operations require special treatment of the periodic condition rather than simply relying on expected changes to cell indices between neighboring cells. We have adjusted this sentence to make this clearer:
"Planar meshes can easily be made periodic by taking advantage of the unstructured mesh specification, such that for most operations periodic cell relationships are handled the same as for neighboring cell relationships."

page 4, line 31: Each core chooses its own vertical coordinate system.

Again, I presume this is "core" used in the sense of MPAS model component but by a computational scientist it could be misinterpreted as CPU-core, also in view of a later occurrence (page 6, line 11) where you actually use "core" in the sense of CPU-core: It is a massively parallel code by design and recently it has been adopting the Kokkos (Edwards et al., 2014a) programming model to provide manycore performance portability (Demeshko et al., 2018) on major HPC platforms

This has been clarified to refer to "MPAS core". I have searched the entire document and used the term "MPAS core" instead of just "core" wherever that is the intended meaning.

page 5, line 18: Additionally, the JIGSAW(GEO) mesh generation tool (Engwirda, 2017a, b) can be used to generate high quality variable resolution meshes with data-based density functions very efficiently.

Please, explain if and how this is relevant to your application. Do you for instance use some Hessian of velocity field in order to manipulate mesh density?

This mention of the meshing tool is included for reference due to the unusual form of the model meshes used by MALI (Voronoi), but applications of the tool form a more detailed topic that are specific to each mesh created and its intended purpose. However, we've added a general statement about applications:
"Density functions that are a function of observed ice velocity or its spatial derivatives and/or distance to the existing or potential future grounding line position have been used."

page 6, line 27. eq. (2):

$$\frac{\partial \sigma_{ij}}{\partial x_i} + \rho g = 0, \quad i,j = 1,2,3$$

Gravity should be displayed as a vector in this balance and hence is missing the index i .

Good catch – this has been corrected.

page 7, line 9: In Equation 5, $A$ is a temperature dependent rate factor, n is an exponent commonly taken as 3 for polycrystalline glacier ice, and γ is an ice "stiffness" factor (inverse enhancement factor) commonly used to account for other impacts on ice rheology, such as impurities or crystal anisotropy (see also Section 6.1).

If you are referring to the commonly applied enhancement factor, $E$, as a pre-factor to the rate factor in Glen's flow law, then not its inverse (as at least I would interpret it), $\gamma = E^{-1}$, but rather $\gamma = E^{-1/n}$ would be correct. This might not be unimportant to spell out, as you use this factor in the inversions and – presumably – plug it back into your forward model.

This is a good distinction to mention. The way we have written gamma here is as it is implemented in the model, so we have clarified this relationship parenthetically to eliminate potential confusion.

page 7, line 18: in which $A_0$ is a constant, $(T^*)$ is the absolute temperature (i.e., corrected for the dependence of melt temperature on ice pressure), $Q_a$ is the activation energy for crystal creep, and $R$ is the gas constant.

I would see $T^*$ (BTW. why do you put it into brackets?) to be (quoting Greve and Blatter, 2007) the temperature relative to the pressure melting point and not the absolute temperature.

That clarification to the definition has been made and the parentheses have been removed.

page 8, line 2: Ice sheets typically have a small aspect ratio, small surface and bed slopes, and vertical pressure distributions that are very nearly hydrostatic.

In the light that you apply Blatter-Pattyn, in my opinion it would be better not to refer to pressure but rather the vertical distribution of the vertical normal Cauchy stress, i.e. the hydrostatic stress approximation (Greve and Blatter, 2007).

Good point. Considering the fact that the validity of the hydrostatic stress approximation is a consequence of the ice sheet small aspect ratio, we decided to mention only that the small aspect ratio and the small surface and bed slopes as a motivation for the reduced-order approximations.

page 9, line 3, equation (15):

If the symbol $\mu$ stands for the effective viscosity (that is what I presume), you are deviating from the symbol $\eta_f$ you used in equation (9) – which by subscript (e -> f) again deviates from its initial definition in equation (5) and also from the definition in the table of symbols - and you should correct it. If it defines something else, then you should declare it. Same accounts for equation (16). For someone who wants to really understand what you are doing in your code, consistency in notation is very helpful.

This variable has been corrected to consistently be $\mu_e$ everywhere.

page 9, line 5, equation (16):

The first term in equation (16) to me reads a scalar-product between two vectors, which leads to a scalar, but the other terms to me read as a with the surface normal aligned vector. To me this seems to be inconsistent.

Thanks for noticing the inconsistent notation, we have fixed it.

page 12, line 26: … $k$ is assumed constant and uniform

The temperature dependent heat conductivity of ice varies about 34% between atmospheric pressure melting point and -50 C (Greve and Blatter, 2007). As this is a temperature span that easily can occur in a single column of the Antarctic ice sheet, could you please elaborate how big of an error do you introduce by using a constant value, and which value (corresponding to which temperature) you are choosing?

We have added some clarifying text to the section explaining this methodological choice:
"The method for evolving ice temperature and default parameter value choices are adapted from the implementation in the Community Ice Sheet Model \citep{price2015}, which is in turn based on the Glimmer model \citep{Rutt2009}. The choice of constant $k$ with a temperate ice value will lead to underestimation of conduction in cold ice. Relaxation of this assumption is planned for future releases of MALI."
The value used (2.1 W/mK) is listed in Table A.1.

page 19, line 17, equation (52):

$$\boldsymbol{\tau_b} = C_0 N \boldsymbol{u_b}^m$$

A minor issue, but this type of notation leaves room for (mis)interpretation of a vector subject to an exponent. Looking at equation (15), I conclude you mean that the exponent applies component-wise. My suggestion: Either mention that in the text or perhaps write (52) out on a component level – just like (15).

This has been corrected.

page 21, line 1: … , suggesting a basal friction law based on the subglacial hydrologic state could be configured to yield realistic ice velocity.

This links to one of the major points of criticism: I do not think you can draw that conclusion from a run done on 20 km grid spacing, when the hydrology in the major outlet systems takes place on subgrid scale.

This statement has been removed. See above for response to general resolution concerns.

page 21, line 10: The calving front is maintained at its initial location by adding or removing ice after thickness evolution is complete. This option does not conserve mass or energy but provides a simple way to maintain a realistic ice shelf extent (e.g., for model spinup).

I don't get the point of this statement: Doesn't this apply to any sort of calving process? In my opinion introduction of discontinuous calving processes in a continuum model, simply by the fact that it instantaneously removes parts of the continuum, by definition drains mass and energy. I guess you referring to the artificially added mass – then it would be interesting to know how you implemented this. Do you add a layer of minimum ice-depth of 1m as on land?

The statement about violation of conservation was referring to the case where thin ice has to be added back in maintain the same ice extent. This has been clarified in the text and we also state that this ice has a 1 m thickness.

page 31, line 10: The Antarctica model configuration we demonstrate here uses a 20 km uniform resolution mesh, ...

As I explained before, I see 20 km resolution as a problem, in particular in connection with the marine ice sheet dynamics. In particular, as you claim yourself on page 29: Thus for marine ice sheets with similar configuration to the MISMIP3d test, we recommend using MALI with the grounding line parameterization and a resolution of 1 km or less.

See general response above regarding improved mesh resolution used in this section.

**Typos and type-setting**

page 7, line 23: Ice sheet models solve Eq. 2-8 with ...

Please, check how GMD wants to have references to equations. I see inconsistencies throughout the text, with either the word Equation/s spelled out or abbreviated (like here). Also check, whether you should put the equation numbers into brackets or not.

We have updated the text to follow GMD's guidelines for Equation, Figure, Table, and Section references.

page 17, line 7: ... space can be represented by the effective water depth in the macorporous sheet, W: macorporous -> macroporous

Fixed.

page35,line17: An implicit subglacial hydrology model basedon existing such models(Werderetal., 2013; Hoffman and Price, 2014) is under development using the Albany framework.

To me this sentence reads strange, except if the word "such" is removed

Here "such" refers to subglacial hydrology models that are implicit, as opposed to the explicit approach presented in the paper.

page 42, line 7: Those longish (and in this case for my browser not existing) links could be dropped - these occur several times throughout the list of references. In general, I think the DOI link is sufficient and no other link is needed. Perhaps GMD has a policy for reference-styles.

The unneeded URLs have been removed from the references.